# RePIC: Reinforced Post-Training for Personalizing Multi-Modal Language Models

**Yeongtak Oh**[1], **Dohyun Chung**[2], **Juhyeon Shin**[3], **Sangha Park**[1],
**Johan Barthelemy**[5], **Jisoo Mok**[4†], **Sungroh Yoon**[1,3†]

[1]Department of Electrical and Computer Engineering, Seoul National University
[2]Department of Future Automotive Mobility, Seoul National University
[3]Interdisciplinary Program in Artificial Intelligence, Seoul National University
[4]Daegu Gyeongbuk Institute of Science and Technology [5]NVIDIA

{dualism9306, pissaitworks, newjh12, wiarae, sryoon}@snu.ac.kr,
jmok@dgist.ac.kr, jbarthelemy@nvidia.com

## Abstract

Recent multi-modal large language models (MLLMs) often struggle to generate personalized image captions, even when trained on high-quality captions. In this work, we observe that such limitations persist in existing post-training-based MLLM personalization methods. Specifically, despite being post-tuned with large-scale caption data through supervised fine-tuning (SFT), these models frequently fail to produce faithful descriptions in real-world scenarios, such as multi-concept image captioning. However, acquiring large-scale, high-quality captions for such complex settings is both costly and difficult. To address the data-centric nature of SFT, we propose a reinforcement learning (RL)-based post-training framework. To the best of our knowledge, this is the first RL-based approach to post-train MLLMs for personalized image captioning. Our method significantly enhances both visual recognition and personalized generation capabilities of MLLMs, and consistently outperforms existing SFT-based baselines, especially in the challenging multi-concept image captioning task. Project page: https://github.com/oyt9306/RePIC

## 1 Introduction

The emergence of Large Language Models (LLMs) has greatly propelled the advancement of AI, particularly in natural language understanding and generation [1, 52, 13, 6]. These models demonstrate impressive general knowledge and achieve strong performance across a wide range of tasks [17, 55]. This progress inspired the development of Multimodal Large Language Models (MLLMs) [31, 25, 57, 4]. MLLMs integrate visual inputs using pretrained vision encoders, treating image embeddings similarly to text tokens within a unified architecture. This integration [31, 30, 4] extends the utility of LLMs to vision-language tasks, enabling image-grounded dialogue [25] and captioning [25, 30, 4].

Unfortunately, despite being pre-trained on large-scale datasets, MLLMs struggle to perform personalization by recognizing and incorporating personal, user-specific concepts, typically provided in the form of a reference image of the user and an associated textual description [40, 15]. Figure 1 illustrates MLLMs' failure in personalized image captioning, one of the most widely-studied tasks in MLLM personalization. Given the reference image of 'thao' and the corresponding description about her, *i.e.,* 'A 23-year-old woman who adores her beloved dog, Bo.', an MLLM is prompted to provide a personalized caption for a new query image of 'thao' that significantly differs

---

†Corresponding authors

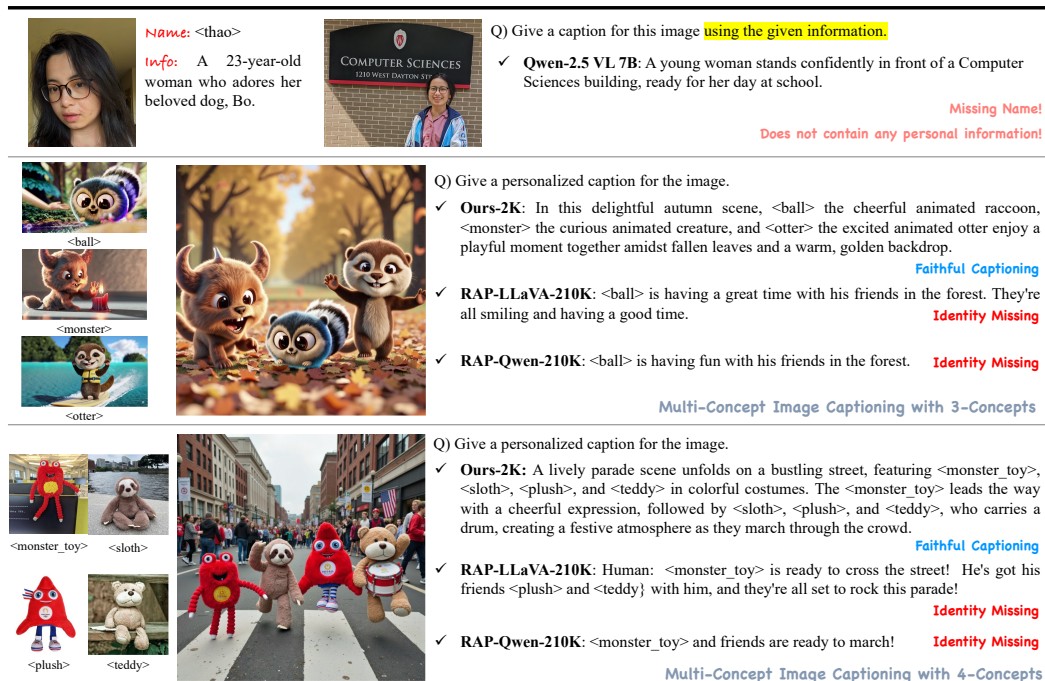

Figure 1: Visualizations of personalized image captioning results. In the first row, the zero-shot MLLM frequently fails to generate personalized captions. The used images are sourced from Yo'LLava [37]. The remaining rows illustrate multi-concept scenarios at inference time. Compared to other SFT-based methods, our approach consistently produces faithful and detailed captions while accurately recognizing all provided identities, even for 3 or 4 concepts. All images are sourced from MuDI [22].

from the reference image in lighting, background conditions, and poses; a successfully personalized caption should accurately refer to 'thao' and include faithful details about the query image. However, Qwen-2.5 VL 7B [4], one of the most performant open-source MLLMs, fails to recognize 'thao' and include her personal concepts in its caption. Thus, recent works [2, 37, 40, 15] have increasingly focused on the personalization of MLLMs.

Existing approaches for MLLM personalization can be categorized into two groups: those that require retraining as new personal concepts are introduced [2, 37], and those that do not [40, 15]. In real-world applications, where the types of personal concepts are unpredictable and new ones continuously emerge, the former family of personalization approaches is inherently highly limited. The latter cases [40, 15], which enable MLLM personalization without the need for retraining when new concepts appear, fine-tune the LLM on large-scale label-annotated datasets composed of question-answering pairs using Supervised Fine-Tuning (SFT). As a result, the post-tuned MLLM can recognize corresponding personal concepts between reference and query images and generate outputs including the given personal information at inference time.

To curate training data composed of image-caption pairs, previous SFT-based approaches [40, 15] have relied on proprietary MLLMs such as GPT-4o [21] and Gemini [51] to generate large-scale personal captions. For instance, PVIT [40] used GPT-4o to create captions and manually validated them for individual human images. Similarly, RAP-MLLM [15] curated captions using Gemini. However, even after post-training with large-scale captions, we observe that existing SFT-based post-tuned MLLMs still often struggle to generate faithful personal captions for the query image.

As shown in Figure 1, these difficulties become more pronounced in real-world scenarios, such as involving 3 or 4 concepts. Following our investigation, the SFT-based method [15] underperforms primarily due to the scarcity of captions in training data for multi-concept settings (*i.e.*, only 5.4% of the total dataset). However, considering the fact that the performance of SFT is highly sensitive to the quality of the training data [7, 48, 32], obtaining a large volume of high-quality personal captions for SFT is both costly and challenging. This challenge becomes particularly severe when

curating captions for images containing multiple distinct identities, as each caption must accurately incorporate detailed personal information corresponding to every identity represented.

To overcome these difficulties, we investigate the key capabilities that a MLLM should possess for personalized image captioning: (1) *robust visual recognition ability*: the ability to consistently identify the same object across different images, even under variations in pose, location, lighting, and background. This ability induces the MLLM to describe the query image accurately and faithfully; and (2) *consistent personalized generation ability*: the ability to incorporate personal information from demonstrations into its responses, including correctly referencing the provided names. MLLM equipped with this capability can perform personalized image captioning.

To strengthen the abovementioned key capabilities, we propose a **Re**inforced post-training for **P**ersonalized **I**mage **C**aptioing (**RePIC**) framework. We leverage the strengths of reinforcement learning (RL) for a method composed of three key components, as outlined below:

**Object Consistency**: To strengthen the MLLM's recognition abilities, we propose an object consistency reward that provides direct positive and negative feedback for the output.

**Visual Localization**: To further reinforce the MLLM's visual recognition ability, we exploit the visual localization reward that predicts bounding box (BBox) coordinates based on a query instruction.

**Identity Consistency**: To enhance the MLLM's ability to generate personalized responses, we introduce an identity consistency reward that explicitly encourages the inclusion of target names in the output.

In our experimental results, we reveal that SFT-based personalization methods are highly limited for visual recognition and generalization abilities. Conversely, by integrating our proposed reward templates along with curated datasets and instructions, our method achieves significant performance improvements over existing baselines, particularly in multi-concept personalized image captioning benchmarks. To the best of our knowledge, this is the first work to present an RL-based post-training framework that enables MLLMs to perform personalized image captioning effectively.

## 2  Related Works

**MLLM Personalization** To enable general-purpose MLLMs [31, 31, 4] to perform personalized image captioning, several methods have been proposed, including those requiring retraining when new personal concepts emerge [2, 37] and those that do not [40, 15]. In the former case, MyVLM [2] uses external concept heads to identify user-specific concepts and learns embeddings for each to input into the LLM. Yo'LLaVA [37] encodes personal concepts as special textual tokens that serve as concept identifiers. However, these methods lack scalability to new concepts, as they require retraining the concept identifiers whenever a new concept emerges and do not guarantee sufficient training data per concept. To overcome these limitations, SFT-based post-training approaches have emerged. PVIT [40] uses special prefixes to encode individual-specific information, enabling MLLMs to answer queries about new individuals. RAP-MLLM [15] presents a pipeline combining retrieval for visual demonstrations and post-training for personalized generation based on the query and retrieved concept information. Compared to existing SFT-based approaches, we propose an RL-based post-training method that reduces reliance on large-scale, high-quality personal captions and demonstrates superior effectiveness, particularly in multi-concept personalized image captioning.

**RL-based MLLM Post-Training Methods.** RL has demonstrated substantial improvements in several tasks of LLMs [14, 46, 60, 36, 11]. Recent studies have extended RL-based post-training to MLLMs—applying preference-based RL for hallucination mitigation [61] and model alignment [27], and policy-based RL for visual reasoning [34, 20, 49]. For example, regarding policy-based RL, Visual-RFT [34] proposes reward templates to improve performance on visual perception tasks such as fine-grained image classification and few-shot object detection. Vision-R1 [20] combines cold-start initialization with RL training to enhance MLLM reasoning capabilities, particularly in tasks like bounding box prediction and few-shot classification. Reason-RFT [49] introduces a two-phase RL framework that integrates SFT-based and RL-based methods for visual reasoning. In contrast to these approaches, we leverage policy-based RL to enhance personalized image captioning in MLLMs.

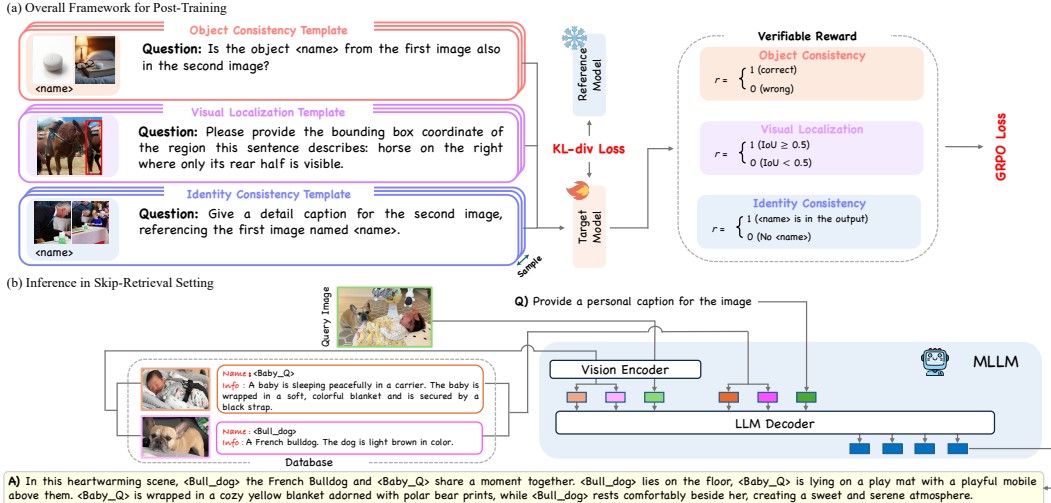

Figure 2: Overview of our RePIC framework: (a) training phase and (b) inference phase. An abbreviated example of the prompt template is shown; complete templates are provided in the Appendix.

## 3 Method

In this section, we detail the GRPO [14] algorithm employed in our approach. To enhance both visual recognition and personalized generation capabilities, we introduce an RL-based post-training framework for MLLMs. Specifically, we outline the design of verifiable rewards and the structuring of instruction and data templates. Figure 2 provides an overview of our proposed RePIC framework during both training and inference stages.

### 3.1 Preliminary: Group Relative Policy Optimization (GRPO)

GRPO [46] is an improved RL algorithm of PPO [44] that exploits the rewards and introduces group-based learning based on relative preferences. Specifically, as shown in Eq. (1), it optimizes a clipped surrogate objective, similar to PPO, while simultaneously minimizing the Kullback-Leibler (KL) Divergence between the current and reference policies. For each state, a task $q$ is sampled from the distribution $\mathcal{Q}$, and the policy model $\pi_{\theta_{\text{old}}}$ generates a set of $G$ responses, denoted as $\{o_i\}_{i=1}^{G}$, which are referred to as rollouts. Then, each response is assigned with a corresponding reward $\{r_i\}_{i=1}^{G}$. This process can be formulated as follows:

$$\mathcal{L}_{\text{GRPO}}(\theta) = \mathbb{E}_{q \sim \mathcal{Q}, \{o_i\}_{i=1}^{G} \sim \pi_{\theta_{\text{old}}}(\cdot|q)}$$

$$\left[ \frac{1}{G} \sum_{i=1}^{G} \frac{1}{|o_i|} \sum_{t=1}^{|o_i|} \min\left( r_t^i(\theta) \hat{A}_t^i, \text{clip}\left(r_t^i(\theta), 1-\epsilon, 1+\epsilon\right) \hat{A}_t^i \right) - \beta \, \mathbb{D}_{\text{KL}}(\pi_\theta \| \pi_{\text{ref}}) \right] \quad (1)$$

where the importance of sampling ratio $r_t^i(\theta) = \frac{\pi_\theta(o_{i,t}|q, o_{i,<t})}{\pi_{\theta_{\text{old}}}(o_{i,t}|q, o_{i,<t})}$, $\hat{A}_t^i$ is the normalized advantage with $\mu$ and $\sigma$ being the mean and standard deviation of a group of rewards. Here, $\beta$ controls the strength of the KL-regularization.

We leverage a *verifiable reward* (VR) [14] with GRPO, enabling training without the need for an auxiliary reward model. In the context of personalization, the following sections discuss how the core capabilities—visual recognition and personalized generation—can be formulated using VRs.

### 3.2 Proposed Verifiable Rewards

**Object Consistency Tuning (OCT)** To improve the visual recognition capabilities of MLLMs by providing a VR, we construct positive-negative pairs. Positive pairs consist of images containing the same object, while negative pairs include images with different objects. We then use binary-response questions, such as ''Is <name> present in the second image?'', and assign a VR of 1 if the

model responds `yes` for a positive pair and `no` for a negative pair. Incorrect responses receive a VR of 0. Based on these pairs, we design a reward template called OCT, as defined in Eq. (2).

$$r_{\text{OCT}} = \begin{cases} 1, & \text{if correctly answer with } \texttt{yes} \text{ or } \texttt{no} \\ 0, & \text{else} \end{cases} \quad (2)$$

We use real datasets such as COCO [29], Objects365 [45], and CelebA [33], from which we crop object regions to serve as reference images. However, as real data often lacks sufficient variation in attributes such as pose and lighting, we additionally incorporate high-quality, visually diverse synthetic images from Subject 200K+ [50]. These synthetic images, generated using diffusion models such as Flux [24], include variations in pose and background while preserving subject identity.

**Visual Localization Tuning (VLT)** To enhance MLLM by strengthening its localization capability, we adopt the IoU-based accuracy reward template introduced in VLM-R1 [47]. In this setup, the Intersection over Union (IoU) score serves as the reward criterion, where a VR is assigned as 1 if the predicted BBox aligns with the GT BBox at an IoU threshold greater than $0.5$, as described in Eq.(3):

$$r_{\text{VLT}} = \begin{cases} 1, & \text{if } \texttt{IoU} \geq 0,5 \\ 0, & \text{otherwise} \end{cases} \quad (3)$$

We refer to this reward template as VLT, and by reinforcing VLT, the model is enabled to localize the object within the image and can understand its relative location, such as right, left, or top. Specifically, we use Refcoco/+/g datasets [35, 59] commonly used for the general visual reasoning task of referring expression comprehension (REC). Notably, our empirical findings indicate that removing the REC task from the training set often leads to instability during RL-based post-training.

**Identity Consistency Tuning (ICT)** To force the model to consistently utilize the provided information from visual demonstrations in its responses, we consider a positive pair composed of a few reference images and a query image pair. For the reward template of using one reference image per query, we call Single-ICT, and for the reward template of using multiple reference images per query, we call Multi-ICT. In this setting, we assign a unique name to each reference image as `<name>` token. In detail, we prompt the model with response questions such as ''`Describe the query image while referencing the reference images.`'', and assign a VR of 1 is only assigned when the model accurately describes the query image using all the given names. Note, in our experiments, we used a maximum of 3 reference images per query image. In detail, for single-ICT, we assign a VR as follows in Eq. (4):

$$r_{\text{Single-ICT}} = \begin{cases} 1, & \text{if } \texttt{<name>} \text{ appears in the output} \\ 0, & \text{otherwise} \end{cases} \quad (4)$$

We use the positive pair images used for OCT. Next, for multi-ICT, for a given set of $m (\leq 3)$ names and $n$ correctly mentioned names, we assign a VR for the response as described in Eq. (5):

$$r_{\text{Multi-ICT}} = \begin{cases} n/m, & \text{if } \texttt{<name 1>}, \texttt{<name 2>}, \cdots, \texttt{<name n>} \text{ appears in the output} \\ 0, & \text{otherwise} \end{cases} \quad (5)$$

To construct multiple reference images, we use real multi-object images from COCO [29] and Objects365 [45], and manually curate high-quality examples by cropping two or three distinct objects from a single query image. The data templates used for each component are provided in the Appendix.

Furthermore, we incorporate descriptive prompts (*e.g.*, '`describe this image in detail.`') that elicit richer language generation in the training dataset. Additionally, we apply output length regularization to ensure that captions exceed a minimum length, which helps avoid less preferable responses (*e.g.*, '`This is <name>.`').

## 4  Experiments

**Baseline** We compare our method with other post-training-based approaches. As baseline models, we fine-tuned PVIT-LLaVA [40] using a 210K subset of the 3M dataset. We also consider pre-trained RAP-LLaVA [15], which is fine-tuned using LoRA [19] from LLaVA-1.5 Vicuna 13B [30]. In addition, we fine-tune RAP-LLaVA using only 2K samples randomly selected from the full

Table 1: Single-concept personal grounding performance evaluation results.

| Models | Seen Data | MyVLM [2] | | | Yo'LLaVA [37] | | | DreamBooth [43] | | |
|---|---|---|---|---|---|---|---|---|---|---|
| | | Pre. | Rec. | F1 | Pre. | Rec. | F1 | Pre. | Rec. | F1 |
| *Skip-Retrieval Setting* | | | | | | | | | | |
| PVIT-LLAVA | 210K | 17.1 | 1.8 | 3.3 | 20.1 | 2.1 | 3.8 | 26.5 | 16.5 | 20.3 |
| RAP-LLAVA | 210K | 100 | 92.9 | 96.3 | 100 | 95.5 | 97.7 | 97.3 | 91.8 | 94.5 |
| RAP-LLAVA | 2K | 100 | 49.4 | 66.1 | 50.6 | 48.6 | 49.6 | 68.4 | 65.8 | 67.1 |
| RAP-Qwen | 210K | **100** | **98.8** | **99.4** | **100** | **99.8** | **99.8** | **100** | **100** | **100** |
| Qwen-2.5 VL | 0 | 100 | 56.8 | 72.4 | 100 | 33.3 | 50.0 | 96.0 | 76.6 | 85.2 |
| Ours | 2K | 100 | 96.2 | 98.1 | 99.7 | 96.1 | 97.9 | 100 | 98.1 | 99.0 |
| *Retrieval Setting* | | | | | | | | | | |
| Retrieval (Top-2) | | 97.6 | 95.9 | 96.7 | 83.6 | 82.9 | 83.3 | 99.3 | 96.2 | 97.7 |
| RAP-LLAVA | 210K | 95.6 | 79.1 | 87.8 | 82.7 | **79.9** | **81.2** | 96.0 | 91.1 | 93.5 |
| RAP-LLAVA | 2K | 79.2 | 53.8 | 64.1 | 71.2 | 52.2 | 64.4 | 69.5 | 66.5 | 68.0 |
| RAP-Qwen | 210K | 95.5 | **87.9** | **91.6** | 79.2 | 75.1 | 76.2 | **98.7** | **94.3** | **96.4** |
| Qwen-2.5 VL | 0 | 91.5 | 50.6 | 65.2 | 77.4 | 42.3 | 55.2 | 95.2 | 75.3 | 84.1 |
| Ours | 2K | **99.0** | 83.2 | 90.4 | **84.4** | 69.7 | 76.3 | 98.6 | 90.5 | 94.4 |

Table 2: Multi-concept personal grounding performance evaluation results.

| Models | Seen Data | 2-Concepts | | | | | | 4-Concepts | | | | | |
|---|---|---|---|---|---|---|---|---|---|---|---|---|---|
| | | Skip-Retrieval | | | Retrieval | | | Skip-Retrieval | | | Retrieval | | |
| | | Pre. | Rec. | F1 | Pre. | Rec. | F1 | Pre. | Rec. | F1 | Pre. | Rec. | F1 |
| RAP-LLaVA | 210K | 100 | 93.9 | 96.9 | 99.3 | 89.6 | 94.5 | 52.9 | 4.3 | 7.9 | 16.7 | 3.1 | 5.2 |
| RAP-LLaVA | 2K | 100 | 90.2 | 94.9 | 95.7 | 81.1 | 87.8 | 36.4 | 1.9 | 3.6 | 22.4 | 0.7 | 1.4 |
| RAP-Qwen | 210K | 100 | 82.9 | 90.7 | **100** | 73.2 | 84.5 | 49.6 | 13.6 | 21.3 | 12.6 | 2.6 | 4.3 |
| Qwen-2.5 VL | 0 | 100 | 75.0 | 85.7 | 98.1 | 64.0 | 77.5 | 73.3 | 22.9 | 34.8 | 22.5 | 6.4 | 10.0 |
| Ours - Full | 2K | **100** | **98.8** | **99.4** | 97.5 | **93.9** | **95.7** | **88.0** | **59.5** | **71.0** | **24.8** | **15.7** | **19.2** |

210K dataset, matching the amount of seen data used in our method. In our study, we adopt the instruction-tuned Qwen2.5-VL 7B [4] as a backbone MLLM. For a fair comparison, we also fine-tune it using LoRA on the 210K instruction dataset [15], with data templates adapted for Qwen-VL compatibility. The resulting model, fine-tuned using LLaMA-Factory [64], is referred to as RAP-Qwen. Hyperparameter sensitivity details are provided in Appendix C.

**Dataset** We consider both single and multi-concept datasets for evaluation. The single-concept data are sourced from Yo'LLaVA, MyVLM, and DreamBooth. These datasets consist of various single-concept images with variations in lighting, pose, and background conditions. For multi-concept evaluation, we use the RAP-MLLM [15] dataset, constructed by collecting YouTube videos and extracting frames that include 2 concepts. To evaluate its personalization capabilities in more challenging scenarios, we experiment on images containing 4-concept cases, never seen during training. To this end, we curate a dataset by crawling images from movie teasers and award ceremonies where multiple celebrities appear together. We select those for our evaluation dataset in which at least 4 distinct concepts are clearly present. Details on the used datasets are provided in the Appendix B.

**Evaluation** We evaluate personalization capabilities under two different settings. First, in the case where GT demonstrations are directly provided at inference time, we refer to this as the skip-retrieval setting. Note that the demonstrations include several reference image-text pairs. Next, we consider the retrieval setting [15] that automates the manual selection of the demonstrations by retrieving the most relevant visual content from a database. Further details are provided in Appendix B.5. Additionally, we newly evaluate personal grounding performance in the skip-retrieval setting using reference images that do not match the query. A lower score in this case implies that the MLLM does not merely duplicate the given demonstrations while performing personalized image captioning. Additional details for the used evaluation templates can be found in Appendix D.

**Implementation Details** For the zero-shot model, we apply a detailed prompt (e.g., ``Output the final answer, including its name in the answer.'') to guide the model to mention the target identity. Note that in multi-concept settings, in-domain (ID) refers to using 2 concepts, while out-of-domain (OOD) refers to using 4 concepts. In the retrieval setting, we retrieve the top-2 most

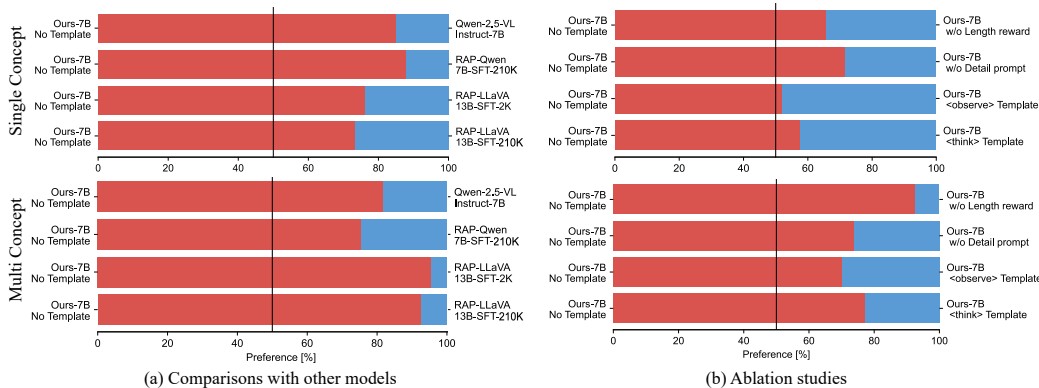

Figure 3: Visualization of preference evaluation scores for single and 2-concept settings, corresponding to the first and second rows, respectively. In (a), our model outperforms all other baseline models, while in (b), it surpasses all ablation variants.

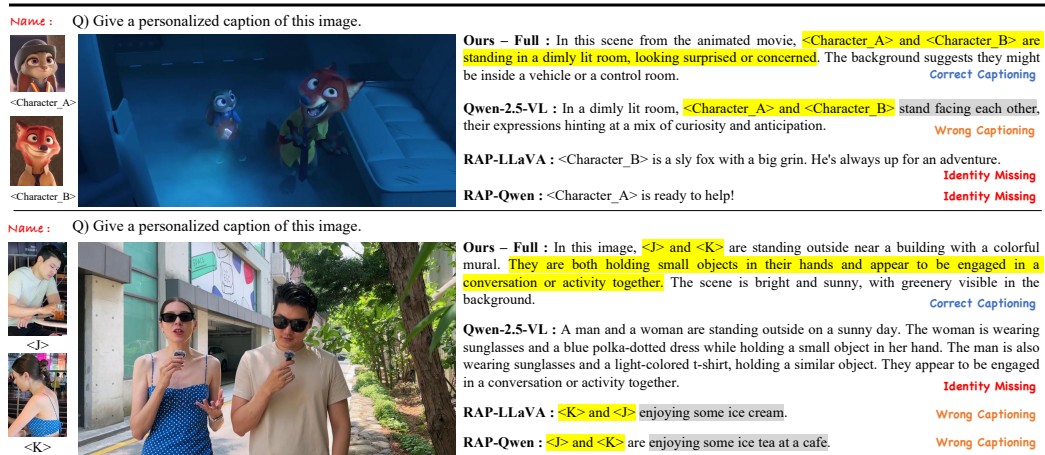

Figure 4: Qualitative examples of 2-concept personalized image captioning.

relevant samples for single and 2-concept settings, and the top-4 for the 4-concept setting. All training experiments are conducted using 8 A40 GPUs, with inference performed on a single A40 GPU. Additional details on the retrieval and experimental setup are provided in the Appendix B.

## 4.1 Personalized Image Captioning Evaluation

### 4.1.1 Personal Grouding Performance

We evaluate model performance for all settings using the same query evaluation prompts. To quantify how frequently the target `<name>` or `Name` or `name` appears in the model's output, we compute precision, recall, and F1-score following the evaluation protocol of [15]. In detail, these scores represent how well the model generates the response while containing the personalized information, without considering the caption quality. We will refer to these scores as the *personal grounding* performance of the post-tuned MLLM. Here, recall reflects the proportion of correctly mentioned target concepts out of total concept names, while precision indicates the fraction of correct mentions out of all concept names that occurred. In the skip-retrieval setting, precision can be reported as 100%, as the GT demonstrations are provided. For multi-concept settings, we assign a score of $n/m$ if the model correctly included $n$ target names in the response from given $m$ names.

As shown in Table 1, in the single-concept setting, the reproduced baseline PVIT [40] shows notably low personal grounding performance, even under the skip-retrieval setting. We conjecture that this poor performance likely stems from the training data, which is largely human-centric. The reproduced

Table 3: Image caption quality evaluation results with reference captions.

| Types | Metrics | RAP-LLaVA | RAP-Qwen | Zero-Shot | Ours |
|---|---|---|---|---|---|
| Reference-based | BLEU [38] $(10^{-2})$ | 0.260 | 0.170 | 0.210 | **0.290** |
| | CIDEr [53] | 0.193 | 0.185 | **0.208** | 0.194 |
| | METEOR [5] | 0.242 | 0.267 | 0.271 | **0.321** |
| | SPICE [3] | **0.104** | 0.084 | 0.083 | 0.086 |
| | BERTScore [63] | **0.683** | 0.567 | 0.523 | 0.668 |

Table 4: Image caption quality evaluation results without reference captions.

| Types | Metrics | RAP-LLaVA | RAP-Qwen | Zero-Shot | Ours |
|---|---|---|---|---|---|
| Reference-free | CLIPScore [18] | 0.332 | 0.316 | 0.323 | **0.339** |
| | ImageReward [56] | -0.094 | 0.087 | **0.287** | 0.130 |

RAP-Qwen, trained on 210K samples, achieves the highest scores in the skip-retrieval setting, while our method performs comparably and outperforms RAP-LLaVA. Under the retrieval setting, the top-performing model varies by dataset. Notably, RAP-LLaVA suffers a significant performance drop when trained on only 2K samples, highlighting the data dependency of SFT. For reference, we also report the top-2 retrieval performance as an upper bound, accounting for retrieval noise.

In the multi-concept settings, as shown in Table 2, the performance of SFT-based methods drops significantly in both 2 and 4-concept scenarios. In contrast, our proposed method consistently and substantially outperforms all SFT-based baselines under both skip-retrieval and retrieval settings. Notably, the performance gap becomes even more pronounced in the 4-concept setting. Note that the zero-shot Qwen outperforms other SFT-based models in terms of personal grounding across all cases, further highlighting the limitations of SFT-based post-training in generalizing to OOD scenarios. These results underscore the effectiveness of our RL-based post-training approach, particularly in extending it to real-world personalized image captioning tasks.

### 4.1.2 Preference Evaluation

We conduct a human-level quality evaluation with GPT-4o [21] to assess the quality of generated personalized image captions. The quality evaluation is conducted as a preference-based assessment, where captions that merely duplicate the provided information or fail to accurately describe the image are considered low-preference. The evaluation template is provided in the Appendix. In Figure 3, we present the preference evaluation on the single-concept YoLLaVA dataset and on the RAP-MLLM dataset, corresponding to the first and second rows, respectively. Those results indicate the superiority of our proposed post-tuning method in generating high-quality and faithful personalized captions compared to all other methods. Our proposed method significantly outperforms (a) all other baselines, and (b) all ablation models, including those without length regularization, without detailed prompts in the dataset, and with reasoning templates.

To better visualize the effectiveness of our method, we present two qualitative examples in Figure 4. In the first row, although the zero-shot model correctly references the given names, its generated captions are less accurate. SFT-based methods fail to recognize identities consistently, often missing given identities, resulting in unfaithful captions. In the second row, while the zero-shot model produces detailed captions, it fails to include the provided names. Conversely, although previous SFT-based methods mention the names correctly, but produce inaccurate descriptions due to limited visual recognition capability. However, in both cases, our method consistently generates faithful and accurate personalized captions for the query images, demonstrating superior personal grounding and visual understanding. Please refer to further qualitative results in Appendix A.

### 4.1.3 Image Caption Quality Evaluations

We conduct both reference-based and reference-free evaluations of image captioning quality on the YoLLaVA dataset. It is important to note that the employed metrics do not assess the degree of personalization achieved by the MLLM; rather, they measure only the overall quality of the generated captions for the given query images. Table 3 presents a metric-based comparison between

Table 5: Personal grounding with wrong visual demonstrations.

| Models | Seen Data | MyVLM [2] (↓) | | | YoLLaVA [37] (↓) | | | DreamBooth [43] (↓) | | |
|---|---|---|---|---|---|---|---|---|---|---|
| | | Pre. | Rec. | F1 | Pre. | Rec. | F1 | Pre. | Rec. | F1 |
| RAP-LLAVA | 210K | 100 | 89.7 | 94.6 | 99.7 | 97.0 | 98.3 | 95.2 | 90.8 | 92.4 |
| RAP-Qwen | 210K | 100 | 69.7 | 82.1 | 98.2 | 82.6 | 89.4 | 85.0 | 71.5 | 77.7 |
| Qwen-2.5 VL | 0 | 100 | 55.6 | 72.4 | 99.0 | **60.4** | 75.0 | 97.2 | 88.6 | 92.7 |
| Ours | 2K | **98.9** | **54.4** | **71.5** | **93.8** | 64.0 | **76.1** | **82.6** | **63.3** | **71.7** |

Table 6: Ablation studies for personal grounding in 2-concept image captioning.

| Models | Seen Data | Skip-Retrieval | | | Retrieval | | |
|---|---|---|---|---|---|---|---|
| | | Pre | Recall | F1 | Pre | Recall | F1 |
| Zero-Shot | | 100 | 75.0 | 85.7 | 98.1 | 64.0 | 77.5 |
| *Reasoning Template Ablations* | | | | | | | |
| Ours `<think>` | 2K | 100 | 90.9 | 95.2 | 97.2 | 84.2 | 90.2 |
| Ours `<observe>` | 2K | 100 | 80.5 | 89.2 | 99.1 | 67.7 | 80.9 |
| *Reward Template Ablations* | | | | | | | |
| Ours w/o ICT | 2K | 100 | 17.1 | 29.2 | **100** | 14.6 | 25.5 |
| Ours w/o OCT | 2K | 99.2 | 73.2 | 84.2 | 99.1 | 67.7 | 80.4 |
| Ours w/o VLT | 2K | 100 | 53.6 | 69.6 | 98.9 | 50.0 | 66.7 |
| Ours only ICT | 2K | 100 | 29.9 | 46.0 | 98.0 | 25.0 | 39.5 |
| Ours only OCT | 2K | 100 | 12.8 | 22.7 | 97.4 | 16.5 | 28.3 |
| Ours only VLT | 2K | 100 | 17.7 | 30.1 | 98.9 | 18.3 | 29.9 |
| *Additional Component Ablations* | | | | | | | |
| Ours w/o length reg. | 2K | 100 | 92.1 | 95.9 | 99.3 | 91.5 | 95.2 |
| Ours w/o detail prompt | 2K | 100 | 86.0 | 92.5 | 98.4 | 74.4 | 84.7 |
| Ours - Full | 2K | **100** | **98.8** | **99.4** | 97.5 | **93.9** | **95.7** |

the generated captions and GT captions (not reference text in the database) for the query image. Since no GT captions are available for the used evaluation dataset, we generated them using GPT-4o by varying the seed three times. As a result, ours show relatively similar results trends, with minor differences in the top 1–2 rankings. In detail, our proposed method achieves the best performance in METEOR and BLEU, the second-best in CIDEr, SPICE, and BERTScore. Table 4 presents the results of image-text alignment evaluation without reference captions. CLIPScore measures the cosine similarity between image and text embeddings, while ImageReward is a general-purpose reward model aligned with human preferences that evaluates image-text alignment quality. Our proposed method achieved the highest CLIPScore and the second-best ImageReward score.

## 4.2 Ablation Studies

**Necessity of RL.** To emphasize the necessity of RL, we examine the limitations of SFT based on our experimental results. First, we analyze the algorithmic shortcomings in visual recognition for personal grounding. Although RAP-Qwen achieves the highest personal grounding scores in the single-concept setting (Table 1), its performance significantly deteriorates when incorrect demonstrations are provided (Table 5). The results suggest that SFT-based methods struggle to distinguish different objects between reference and query images, often resulting in inaccurate personalized captions. In contrast, our RL-based method consistently yields lower scores when provided with incorrect demonstrations, highlighting its robustness in differentiating objects and its ability to avoid merely duplicating the given personal information when generating personalized captions.

Furthermore, we analyze why SFT-based methods struggle with personal grounding in multi-concept settings, as shown in Table 2. To this end, we examine the training data compositions that include multiple identities within a single image. Interestingly, despite a comparable proportion of multiple-identity training data (*i.e.*, RAP-MLLM: 5.4% of 210K, Ours: 4.7% of 2K), SFT yields only marginal improvements in the 2-concept setting (ID) and performs poorly in the 4-concept setting (OOD). In contrast, our method achieves substantial performance gains in both the 2 and 4-concept personalized image captioning tasks.

We attribute the shortcomings of SFT-based methods—particularly their limited visual recognition and poor generalization to OOD scenarios—to fundamental algorithmic differences. As a data-centric approach, SFT tends to overfit to dominant patterns and struggles to learn from rare or diverse

data [12]. Our experimental results further highlight why RAP-Qwen performs well in single-concept settings but fails significantly in OOD scenarios. These findings underscore the necessity of RL-based approaches to perform robust and generalizable personalized image captioning.

**Efficacy of Each Component.** The overall ablation studies are shown in Table 6. First, we evaluate the effectiveness of reasoning templates [14], such as the `<think>` token used in visual reasoning tasks. As a result, using reasoning templates rather degrades the personal grounding performance, regardless of the two different special reasoning tokens. For reward template ablations, removing the OCT template from the training dataset weakens the scores, underscoring the importance of reinforced visual recognition. Most notably, removing ICT causes a drastic decline in performance, confirming its critical role in personal grounding. Removing the VLT also leads to a performance drop, suggesting that reinforcing visual localization contributes to enhancing personal grounding. These findings emphasize that the integration of all proposed components is essential to achieve state-of-the-art performance. For the ablation results on the additional components related to preserving the captioning quality, excluding length regularization or detailed query prompts results in minor accuracy drops. For the qualitative effects of these components on image captioning quality, please refer to the Appendix.

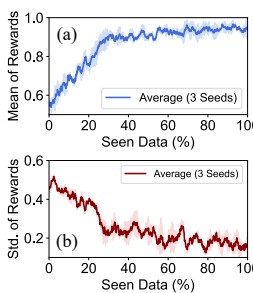

Figure 5: Visualization of training stability.

**Training Stability.** In Figure 5, we report the mean and standard deviation of rewards across different random seeds to demonstrate the stability of our RL training, independent of the data curriculum.

## 5  Conclusion

In this work, we propose RePIC, a novel RL-based post-training framework that alleviates the cost of collecting high-quality personal captions, enabling effective personalized image captioning. By leveraging verifiable rewards, curated data compositions, and tailored instruction templates, our post-tuned MLLM achieves strong performance on personalized captioning tasks across both single and multi-concept settings. Furthermore, our experimental results demonstrate that RePIC equips MLLMs with more robust visual recognition abilities, enhanced personalized generation, and improved generalizability than existing SFT-based methods, highlighting the superiority of our approach in complex scenarios such as 4-concept image captioning cases not seen during post-training.

**Limitations**. In this work, the reproduced baseline models may not have undergone exhaustive hyperparameter tuning. Further, as this study primarily focuses on evaluating the impact of RL on personalized image captioning, future work could extend this approach to other personalized tasks, such as multi-turn conversations. Additionally, we evaluated caption faithfulness only using human-level preference scores due to the lack of GT captions. This would be improved by generating GT captions with larger models like GPT-4o and human refinement.

**Future Work** Reducing retrieval noise through reasoning or stronger retrieval models, and exploring alternatives to length regularization (e.g., self-correction, test-time scaling) are potential directions for future research. While this work focuses on RL-based post-training on the image domain, future research could extend personalization to other modalities such as audio and video, aligned with the progress of models like Qwen-2.5 Omni [57].

## Acknowledgments

The authors also gratefully acknowledge the support from the NVIDIA Academic Grant Program. This research was supported by the National Research Foundation of Korea (NRF) through a grant funded by the Korean government (MSIT) [No. 2022R1A3B1077720]; the Institute of Information & Communications Technology Planning & Evaluation (IITP) grant funded by the Korean government (MSIT) [No. RS2021-II211343], Artificial Intelligence Graduate School Program (Seoul National University), RS-2022-II220959, No.RS-2025-02263754, Human-Centric Embodied AI Agents with Autonomous Decision-Making; Mobile eXperience(MX) Business, Samsung Electronics Co., Ltd.; the AI-Bio Research Grant through Seoul National University; and the BK21 FOUR program, Education and Research Program for Future ICT Pioneers at Seoul National University, in 2025. This research was also conducted as part of the Sovereign AI Foundation Model Project (Data Track),

organized by the Ministry of Science and ICT (MSIT) and supported by the National Information Society Agency (NIA), S.Korea (2025-AI Data-wi43).

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

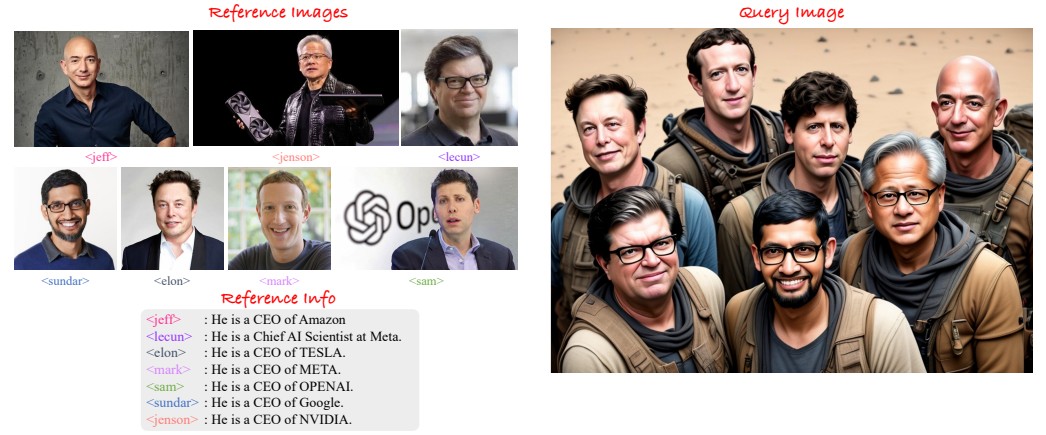

**Q)** Give a detailed personalized caption of the image.

**A)** The query image features a group of influential tech leaders, including <jeff> (CEO of Amazon), <jenson> (CEO of NVIDIA), <lecun> (Chief AI Scientist at Meta), <sundar> (CEO of Google), <elon> (CEO of TESLA),  (CEO of META), and <sam> (CEO of OpenAI). They are standing together in a desert-like setting, wearing tactical gear, suggesting a collaborative effort or a strategic meeting. Their expressions range from serious to smiling, indicating a mix of professionalism and camaraderie among these tech giants.

Figure A.1: Visualization of a more complex and diverse multi-concept image captioning result using our proposed RePIC. The generated query image is sourced from InstantFamily [23].

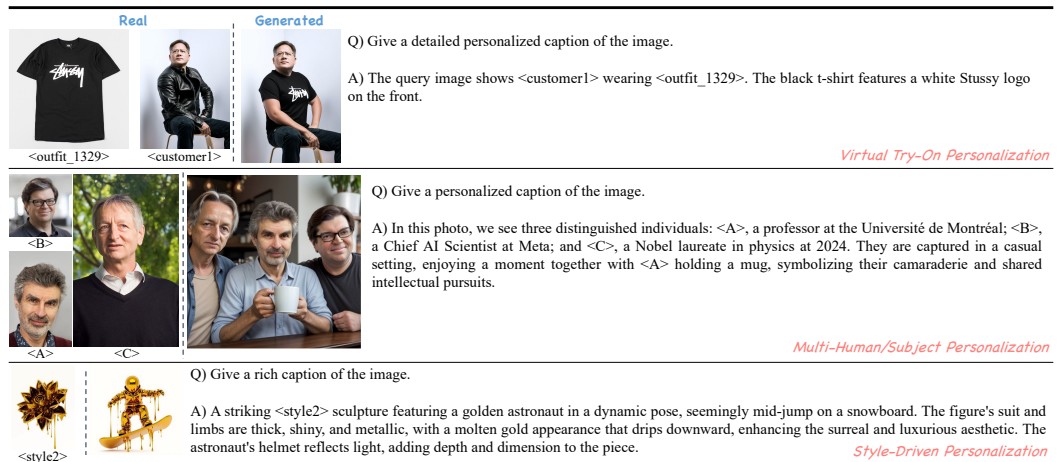

Figure A.2: Visualization of RePIC results on various personalized image captioning tasks.

## A  Additional Qualitative Results

### A.1  More Challenging Multi-Concept Setting

In this section, we further present the results of our method in the 7-concept setting, as illustrated in Figure A.1. Notably, our approach generates faithful descriptions of the query image while accurately referencing the corresponding reference images and their associated information. In Figure A.2, we demonstrate that our proposed method can faithfully perform image captioning on synthetic images across various tasks, including virtual try-on [10], multi-human or subject personalization [16], and style-driven personalization [9], which is compatible with the state-of-the-art personalized image generation benchmarks. These results highlight the superiority of our approach in handling diverse personalized image captioning tasks with MLLM.

### A.2  Visualizations on Image Captioning Quality

In Figure A.3, we present qualitative comparisons of image captioning quality in both single and multi-concept settings, highlighting the effectiveness of our proposed method. Note that the RAP-

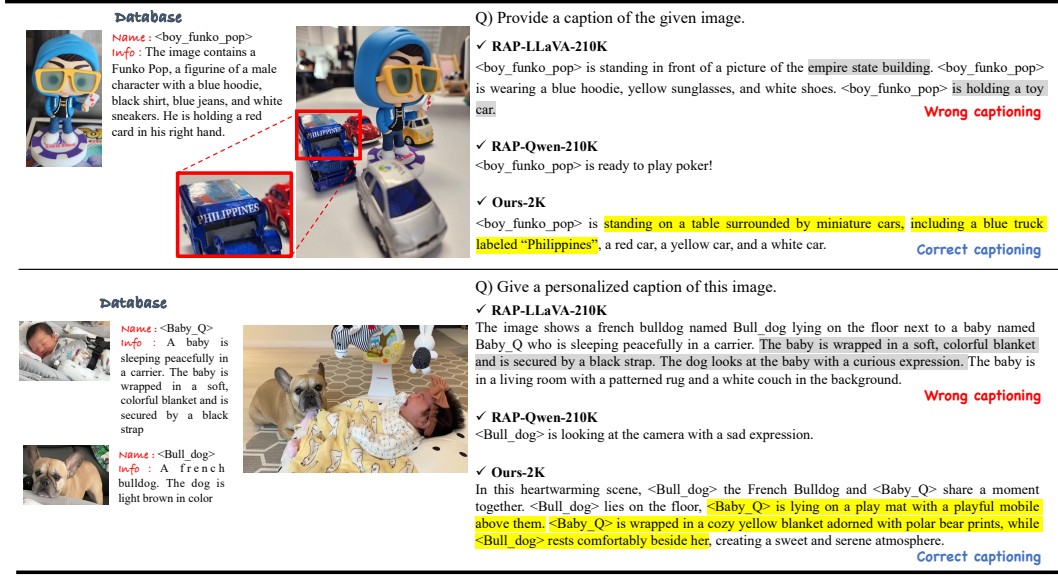

Figure A.3: Examples of generated captions on single and 2-concept personalized image captioning tasks.

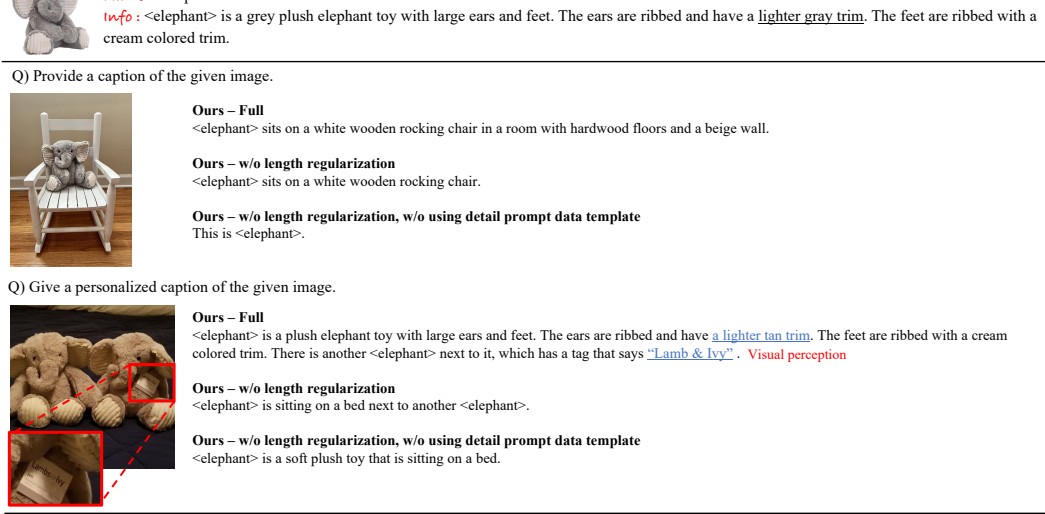

Figure A.4: Visualization of qualitative results for additional components used for our methods.

LLaVA often merely duplicates the retrieval information or generates visual hallucinations without considering vision perception, and RAP-Qwen severely fails to caption correctly. In contrast, only the proposed approach faithfully and concretely describes the given query image.

## A.3 Effects of Additional Components for Preserving Captioning Quality

In Figure A.4, we present additional ablation results evaluating the impact of applying length regularization and incorporating detailed prompts in the training dataset. The results indicate that incorporating length regularization and detailed prompts effectively mitigates generating uninformative captions for the query images.

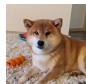

**Name** : \<bo\>

**Info** : \<bo\> is a well-groomed, medium-sized Shiba Inu with a thick, cinnamon-colored coat, cream accents, alert eyes, and a black collar.

Q) Please give a caption of the image.

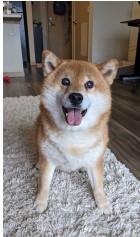

**1) \<think\> template**

\<think\> The image shows a dog that appears to be a Shiba Inu based on its physical characteristics such as the pointed ears, compact body, and facial features. The dog has a thick, cinnamon-colored coat with cream accents, which matches the description of \<bo\>. The dog's expression and posture also align with the description provided. \</think\>

\<answer\> *A well-groomed, medium-sized Shiba Inu* named \<bo\> is sitting on a fluffy rug, looking alert and happy. *The dog's thick, cinnamon-colored coat and cream accents are clearly visible, matching the description of \<bo\>* \</answer\>

**2) \<observe\> template**

\<observe\> The image shows a Shiba Inu dog lying on a white shaggy rug indoors. The dog has a thick, cinnamon-colored coat with cream accents around its chest and paws. Its ears are perked up, and it has a friendly expression with its tongue slightly out. The background includes a wooden floor and some household items like a toy and a blue object. \</observe\>

\<answer\> *A well-groomed, medium-sized Shiba Inu named \<bo\> is lying on a white shaggy rug indoors, displaying its thick, cinnamon-colored coat with cream accents, alert eyes, and a black collar.* The dog's ears are perked up, and it has a friendly expression with its tongue slightly out. The background includes a wooden floor and some household items like a toy and a blue object. \</answer\>

**3) No template**

\<bo\> is sitting on a white rug with its tongue out, looking happy and relaxed.

Figure A.5: Visualizations of output responses with and without the use of reasoning templates.

## A.4 Effect of Reasoning Templates

We consider the post-tuned model trained with reasoning templates such as `<think>` and `<observe>` to verify the effectiveness of visual reasoning in personalized tasks, which has become a prevalent choice [34, 20, 47] for MLLM post-training with RL. However, in Figure A.5, we observe that using reasoning templates often results in longer outputs that fail to faithfully describe the query image. In contrast, omitting templates leads to more concise yet accurate and faithful image descriptions. Furthermore, as demonstrated in our main analysis, reasoning templates negatively impact personal grounding performance in personalized image captioning tasks.

## A.5 Further Limitations of RePIC

We illustrate the limitations of our RePIC model on the personalized image captioning task in Figure A.6. In the first row, RePIC incorrectly captions the image with `blue jeans`, despite no such item being present. A similar issue is observed in the second row, where the model references a `polka-dotted dress` that does not appear in the query image. These examples show a limitation of RePIC in generating accurate personalized captions, primarily due to insufficient fine-grained visual perception. For instance, it struggles when objects are not visibly present (*e.g.*, no blue jeans appear) or when the reference and query images differ significantly (*e.g.*, back view vs. front view), making it difficult to recognize them as the same person or the same object. We expect that these limitations can be mitigated either by constructing a high-quality database for each concept—avoiding the use of personal information based solely on the visual appearance of the image, and ensuring the reference image clearly shows a front view of the object—or by leveraging an MLLM equipped with an advanced vision encoder and a more powerful backbone LLM, such as Qwen-3 [58].

We further acknowledge the inherent limitations of our current personalized retrieval pipeline, particularly in corner cases where relevant concepts in the database are absent from the query image. While our evaluation setting focused solely on object-centric benchmarks and did not account for such cases, we anticipate that addressing them will be an important direction for future work. It is expected to be particularly important in personalization scenarios that use video as input data. In this regard, we believe robustness could be enhanced by incorporating a verifiable reward that penalizes the model for generating responses based on incorrect or irrelevant information. We also conjecture that such scenarios more clearly expose the limitations of RePIC in handling real-world complexity.

# B   Additional Experimental Configurations

## B.1 Experimental Details

Our implementation is based on the open-source codebase. To train our model, we set LoRA rank as 64, LoRA alpha as 128, and use the number of generations per prompt as 8. The base model we used

https://github.com/om-ai-lab/VLM-R1

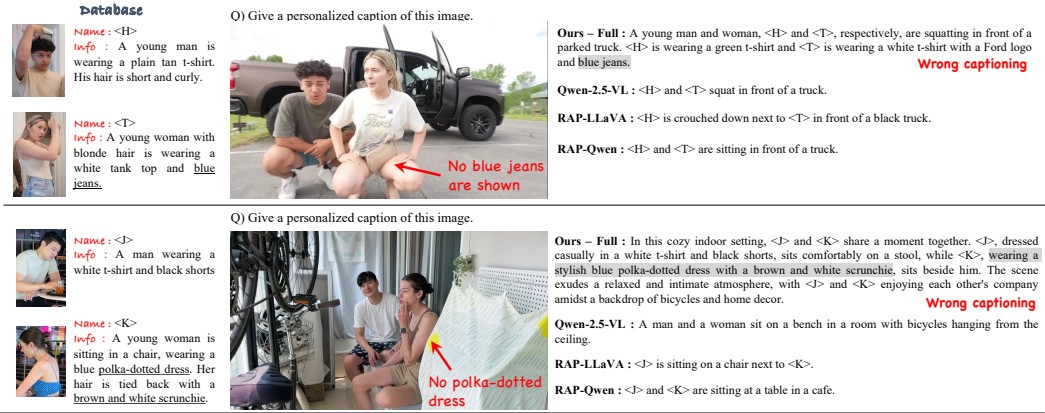

Figure A.6: Examples illustrating additional limitations of RePIC in 2-concept scenario.

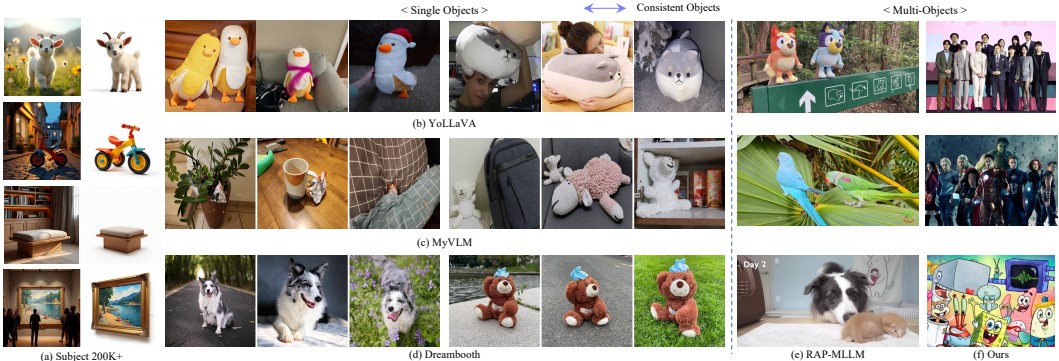

Figure A.7: Datasets used for training and evaluation. Note that the Subject200K+ dataset (a) was used for training, while all real datasets (b) to (f) were used only for evaluation.

is `Qwen2.5-VL-Instruct-7B`, a vision-language model capable of processing images, text, and videos simultaneously, with enhanced instruction-following capabilities. Following the FLAN [54] paradigm, the pretrained MLLM on a large corpus was further fine-tuned using instruction datasets. We adopted this backbone because it is open-source and supports multi-image understanding, a crucial feature for MLLM personalization. We chose the 7B variant as a suitable backbone MLLM for our experiments, and the frozen copy remains fixed throughout training to perform as a reference policy while GRPO.

## B.2 Used Datasets for Evaluation

The data configuration used for both training and evaluation in our experiments is detailed in Figure A.7. Notably, the Subject200K+ dataset was used exclusively for post-training and was not included in the evaluation. All other real-image benchmarks were used for evaluation purposes. In Figure A.8, we present the configuration of our curated DreamBooth [43] database used for single-concept captioning evaluation in our experiment.

## B.3 Used Data Templates

In Figure A.9, we illustrate the data and instructions for verifiable rewards of OCT, VLT, and ICT used for post-training. Table A.1 shows the two reasoning templates of using `<observe>` and `<think>` tokens used for our ablation studies involving special tokens. These templates were appended directly before each captioning query.

< Proposed DreamBooth Database >

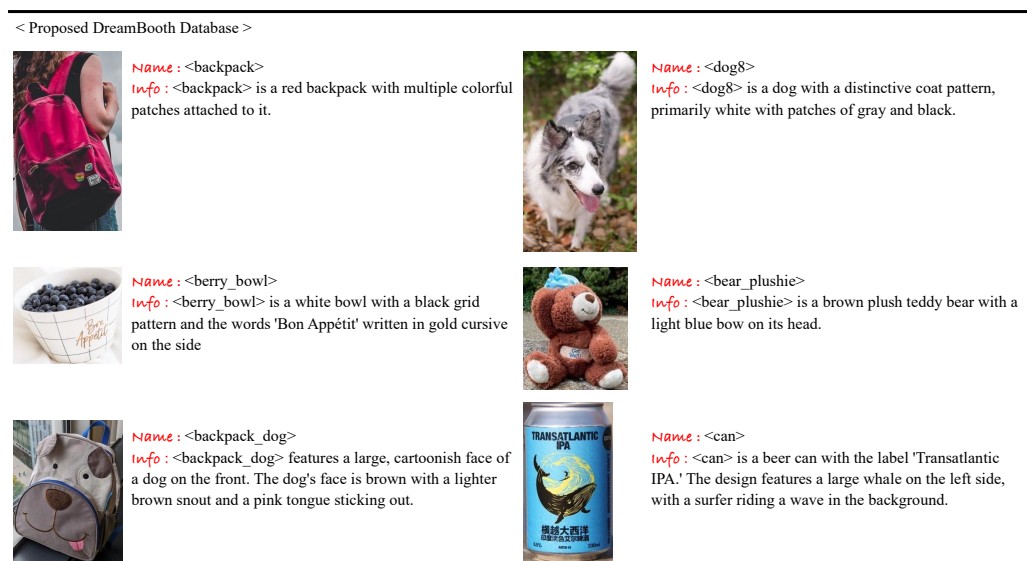

Figure A.8: Visualization of the DreamBooth database constructed in this work.

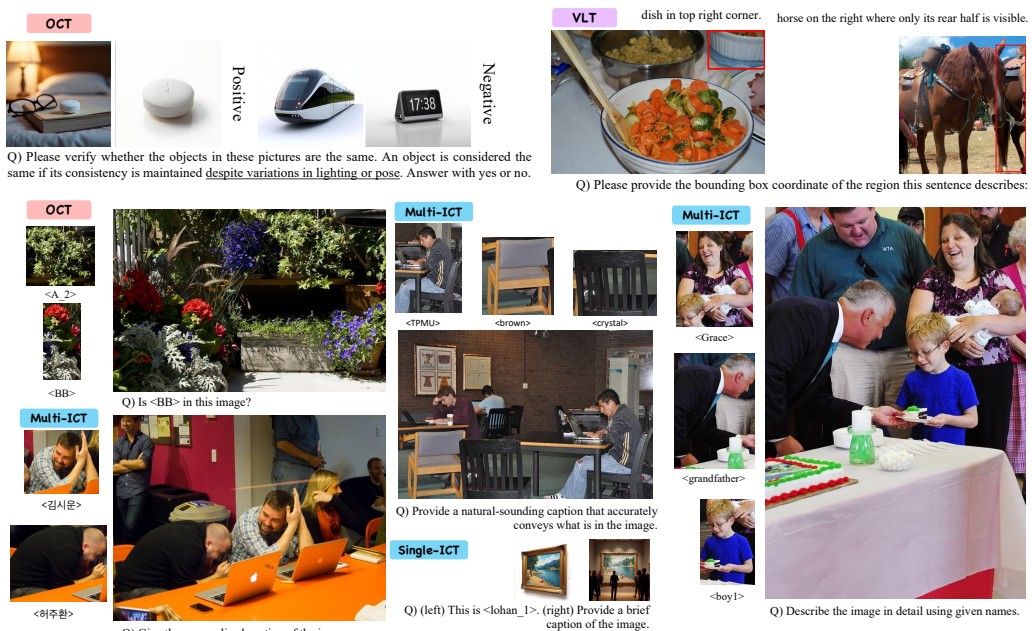

Figure A.9: Visualization of data templates used for MLLM post-training, including examples of OCT, VLT, single-ICT, and multi-ICT.

## B.4 Dataset Compositions

For generating `<name>`, we use a random name generator to sample human or object names in an on-the-fly manner. To be specific, we use the `faker` library to generate multilingual terms (e.g., person and object names) in French, Korean, Italian, Chinese, and English in an on-the-fly manner.

To construct multi-concept crop images, we first select multi-image training samples (approximately 5% of training data) containing multiple crops from RAP-MLLM, and then manually select only those examples with non-overlapping crops and clearly visible identities. These cropped samples

---

https://faker.readthedocs.io/en/master/

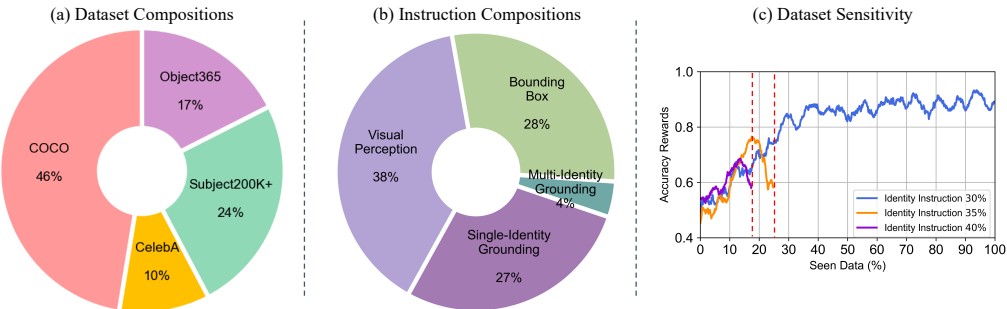

Figure A.10: (a) Dataset composition, (b) instruction composition, and (c) the sensitivity to the proportion of identity grounding instructions within the overall training set.

Table A.1: Used reasoning templates

<think> **Reasoning Template:**
- First output the thinking process in <think> </think> tags and then output the final answer in <answer> </answer> tags.

<observation> **Reasoning Template:**
- First, observe carefully and enclose the observation process in <observe> </observe> tags and then output the final answer in <answer> </answer> tags.

are used without applying additional data augmentation, such as color jittering or rotation. Further, we use the Subject200K+ [50] dataset, which provides pairwise high-quality synthetic images with realistic and diverse lighting and pose variations. By incorporating such a synthetic dataset, we intended to enhance the MLLM's visual perception capabilities for handling more complex and realistic applications, such as personalized AI-generated image captioning.

Figure A.10 illustrates how we construct a high-quality dataset for personal grounding. In (a), we show that the dataset is composed of COCO, Objects365, CelebA, and Subject200K+. (b) visualizes the instruction composition, which includes OCT, VLT, single-ICT, and multi-ICT. We note that approximately 31% of the total training data is composed of single and multi-ICT samples. In (c), we highlight the dataset sensitivity for convergence. We observe that if the amount of ICT instruction in training data is too high, the RL training often fails. This demonstrates that while RL is inherently data-efficient, it is sensitive to data quality. Despite this, we are the first to empirically identify a stable "sweet spot" in data and instruction composition and reward design that effectively balances and stabilizes the training process, which we rigorously validate in our experiments through both in-distribution and out-of-distribution experiments. Overall, our findings highlight the importance of a well-structured instruction dataset for effective RL-based post-training in MLLM personalization.

## B.5 Details On Retrieval Setting

Following the previous work [15], for a retrieval setting, we first utilize a database of image, text pairs representing user-specific concepts. Given database images, each image is first processed using a pre-trained CLIP [42] encoder to obtain visual embeddings. Then, for a given query image and its corresponding textual instructions, YOLO-World [8] is employed to detect regions of interest. Thus, cropped images are encoded into embeddings, and by computing Euclidean distances between these embeddings and pre-stored embeddings, the most relevant reference images are retrieved from the database. Our database is composed of key-value pairs for all concepts, where each concept is associated with an image and corresponding textual information. Then, similar to skip-retrieval, the reference image is retrieved from the database. A detector first generates region proposals from the

query image, and then reference images are retrieved with their texts by selecting the top-K samples from the database with high CLIP image cosine similarities.

More specifically, a detector generates ROIs for the query image relevant to the predefined category texts specified for each concept, like "person", "dog", "toy", and so on. Then, the CLIP similarities are computed between multiple ROIs captured from the query image and the reference images in the database. Specifically, for each ROI generated by the detector, we extract its visual embedding using the CLIP image encoder and compute cosine similarity against the image embeddings of candidate reference images, also obtained using the same CLIP model. The top-K most similar reference images and their texts are then selected for MLLM demonstration. Note, we do not use reference texts in the database for the retrieval. Once the reference images and texts are retrieved, they are prepended to the query prompt in an in-context learning (ICL) fashion. Importantly, no example answers for the query image are provided; MLLMs generate responses solely based on the retrieved references (images and texts) and the query image in a personalized manner.

### B.6  Details on Length Regularization

The output length regularization was applied exclusively to the ICT reward, not to OCT or VLT. Further, the ICT reward is granted only when the output exceeds the cutoff length. As discussed in Figure A.4, we observed that naively maximizing the ICT reward could lead the model to exploit trivial shortcuts, such as repeatedly generating phrases like "This is <name>." Since such outputs still receive a full reward of 1, this results in a reward hacking issue. To address this, we introduced a regularization strategy: if the generated output is shorter than a predefined cutoff length, the reward is set to 0, even if the name is correctly included. We empirically set this minimum length to 100, which corresponds to the lower bound of text lengths in our database. Figure A.17(b) presents a qualitative ablation across different thresholds, and we found that a value of 100 offers stable and meaningful reward signals during training. Additionally, to better encourage the model to produce more descriptive captions, we incorporated detailed prompts into the training data (e.g., "Describe the image in detail"), which promotes longer and more informative outputs. Importantly, the model's captioning and instruction-following abilities are preserved through the KL divergence term in the GRPO loss.

## C  Additional Analyses

### C.1  Theorical Analysis of GRPO with PPO

We provide a theoretical comparison between PPO and GRPO to explain why GRPO is a suitable algorithm for RL training in our work. Proximal Policy Optimization (PPO) [44] is a policy-based RL algorithm that stabilizes policy updates by employing a clipping mechanism, as represented in the equation below.

$$L^{\text{PPO}}(\theta) = \mathbb{E}_{(s,a) \sim \pi_{ref}} \left[ \min \left( r_t(\theta) \hat{A}_t, \text{clip}(r_t(\theta), 1 - \epsilon, 1 + \epsilon) \hat{A}_t \right) \right] \quad \text{(A.1)}$$

$r_t^i(\theta) = \frac{\pi_\theta(a_t|s_t)}{\pi_{ref}(a_t|s_t)}$ s the probability ratio between the new policy $\pi_\theta$ and the old policy $\pi_{ref}$. $\epsilon$ is a small hyperparameter that controls the clipping range. However, PPO heavily relies on absolute reward values, making it sensitive to noise or suboptimal design, and its use of a separate value function typically as large as the policy model, for advantage estimation adds significant memory and computational overhead.

In contrast, GRPO is formulated as a group-level extension of PPO, incorporating per-token policy ratios, clipped advantages, and explicit KL regularization, as described in Eq. (1) of our paper. Since rewards are applied per sample, GRPO performs better with sparse reward data and stably learns from relatively better responses rather than relying on absolute reward values.

In our work, GRPO offers the following advantages:

1. *Effectively handling multiple identity references*: For each query, the model generates multiple responses and takes the most preferred responses at a group level.

Table A.2: Single-concept copy-and-Paste analysis on wrong textual demonstrations.

| Models | BLEU ($\downarrow$) | ROUGE-L ($\downarrow$) | METEOR ($\downarrow$) | SPICE ($\downarrow$) | BERTScore ($\downarrow$) |
|---|---|---|---|---|---|
| RAP-LLaVA | 0.415 | 0.897 | 0.544 | 0.510 | 0.713 |
| RAP-Qwen | 0.190 | 0.834 | 0.361 | 0.375 | 0.447 |
| Zero-Shot | 0.129 | 0.677 | 0.276 | 0.261 | 0.427 |
| **Ours** | **0.079** | **0.578** | **0.213** | **0.140** | **0.340** |

2. *Preserving caption quality during personalization*: Instead of relying on a clipping mechanism, GRPO explicitly applies KL divergence regularization, which helps maintain instruction-following capabilities throughout the personalization process.

3. *Robust training under sparse reward conditions*: These strengths are particularly evident in single- and multi-concept ICT tasks during post-training. In the early stages, when the MLLM lacks personal grounding ability, rewards tend to be sparse and hard to learn. However, we observed that as training progresses, the personal grounding ability begins to emerge, and the reward signal becomes richer and more informative.

Therefore, from those perspectives, we adopt the GRPO algorithm as a suitable strategy for our RL-based post-training because of its stability and effectiveness, even when training with a small amount of data (2K).

## C.2 Additional Technical Depth of GRPO

We provide additional explanations as follows to enhance the clarity of using GRPO. In MLLM post-training using GRPO, the vision encoder remains frozen, and only the backbone LLM is fine-tuned. This ensures that the visual representation (extracted by the frozen encoder) stays stable, while GRPO updates only the language model's parameters based on RL signals.

1. *How are group responses generated?*: For each training instance, the model generates a group of G responses $\{o_i\}_{i=1,\cdots,G}$ by sampling from the reference policy $\pi_\theta$. In our work, group responses consist of personalized captions for ICT, predicted bounding boxes for VLT, and binary yes/no responses for OCT. These responses are then converted into verifiable rewards to calculate group-relative advantages.

2. *How are group-relative advantages normalized, and what are their merits?*: For a given set of scalar verifiable rewards composed of ICT, OCT, and VLT, the rewards are grouped and normalized together to calculate the advantage. The formulation of the group advantage provides a stable reward signal, even in low-resource or noisy reward settings. In our work, this merit becomes prominent in single and multi-ICT rewards during post-training. In the early stages, the initial MLLM lacks personal grounding capability, resulting in sparse rewards. However, as training progresses, the rewards become richer, and we observe the emergence of personal grounding ability (see Figure A.12). We pose that this emergence is encouraged by the GRPO on learning from relatively better responses rather than relying on absolute rewards, thereby improving sample efficiency and training stability.

3. *How is reference policy established?*: In GRPO, the reference policy is typically established as a frozen copy of the initial pretrained MLLM, denoted as $\pi_{ref}$, and remains fixed throughout training. This frozen policy serves as a stable anchor point for regularizing the current policy $\pi_\theta$ via an explicit KL divergence penalty, which helps prevent policy drift.

## C.3 Copy-And-Paste Analysis

We assume that current MLLMs often overlook visual information and rely solely on textual cues without a thorough understanding. To address this issue, we conducted the following experiments. The metrics employed are commonly used in image captioning evaluation, such as BLEU [38], ROUGE-L [28], METEOR [5], SPICE [3], BERTScore [63], and measure the similarity between the generated output and the reference text, skip-retrieved from the database. In this context, a higher score indicates greater similarity between the two sentences, which implies a higher degree of copy-and-paste behavior.

Table A.3: Multi-concept copy-and-paste analysis on wrong textual demonstrations.

| Models | BLEU (↓) | ROUGE-L (↓) | METEOR (↓) | SPICE (↓) | BERTScore (↓) |
|---|---|---|---|---|---|
| RAP-LLAVA | 0.143 | 0.649 | 0.284 | 0.335 | 0.404 |
| RAP-Qwen | 0.012 | 0.259 | 0.110 | 0.025 | 0.195 |
| Qwen-2.5 VL | **0.008** | **0.080** | **0.028** | **0.014** | **0.179** |
| **Ours** | 0.015 | 0.240 | 0.093 | 0.034 | 0.188 |

Table A.4: Comparison of text diversity under different query prompt designs.

| Query Prompt | Models | Self-BLEU (↓) | Distinct-1 (↑) | Distinct-2 (↑) |
|---|---|---|---|---|
| **w/o** detailed prompt | RAP-LLaVA | 0.444 | 0.486 | 0.677 |
| | RAP-Qwen | 0.521 | 0.383 | 0.588 |
| | Zero-Shot | 0.409 | 0.613 | 0.542 |
| | Ours | **0.271** | **0.539** | **0.803** |
| **w/** detailed prompt | RAP-LLaVA | 0.324 | 0.483 | 0.730 |
| | RAP-Qwen | 0.137 | 0.523 | 0.815 |
| | Zero-Shot | 0.359 | 0.542 | 0.782 |
| | Ours | **0.128** | **0.624** | **0.850** |

In Table A.2, we investigate the extent of copy-and-paste behavior of MLLMs. In detail, we use skip-retrieval of reference texts, which corresponds to the setting where wrong textual demonstrations are intentionally provided with the reference image. To elaborate, Table A.2 evaluates the extent to which the output of RePIC overlaps with the incorrect information given in the reference texts in a single-concept setting. In other words, it assesses whether RePIC copies the misleading text verbatim rather than grounding its description in the actual visual content when the demonstration contains information that contradicts the image. As a notable result, despite being fine-tuned on large-scale datasets, SFT-based methods are more vulnerable to the copy-and-paste problem than the zero-shot baseline. In contrast, our RL-based method consistently achieves significantly better performance across all metrics. This suggests that, unlike other SFT-based approaches that often produce templated responses without thoroughly recognizing the query image, our proposed RePIC method enables the post-tuned MLLM to generate personalized captions grounded in proper visual understanding of the query image.

In Table A.3, we present the copy-and-paste results for a multi-concept setting using the RAP-MLLM dataset. In this experiment, we feed incorrect textual demonstrations for both concepts. Our method achieves the second-best performance after the zero-shot baseline and shows consistently lower copy-and-paste behavior compared to other SFT-based models. It is important to note that these metrics do not reflect image captioning quality. Upon analyzing the outputs, we observed that the zero-shot baseline, despite producing the lowest overlap scores, frequently fails to generate meaningful captions and instead outputs only its names or irrelevant responses in this specific scenario.

To further examine copy-and-paste behavior without considering the textual attack scenario, in Table A.4, we measured the textual diversity between output responses generated for each concept. Here, Self-BLEU [65] treats each caption as a hypothesis and the others as references, computing BLEU scores accordingly. Distinct-n [26] calculates the ratio of unique n-grams to the total number of generated tokens, offering an intuitive measure of lexical diversity. In this context, low diversity, indicated by high S-B and low D scores, suggests that the generated captions for a given concept are similar to each other, implying copy-and-paste behavior. Conversely, lower S-B and higher D scores indicate greater diversity in generated captions for each concept. Note that the "detailed prompt" refers to appending the phrase ``output the response without duplicating the given information'' directly to the system prompt. As shown in Table A.4, our proposed method consistently produces more diverse personalized captions compared to other methods in all settings. These results further reinforce our main claims that RePIC demonstrates strong robustness in distinguishing between objects and effectively avoids simply replicating the provided information when generating personalized captions.

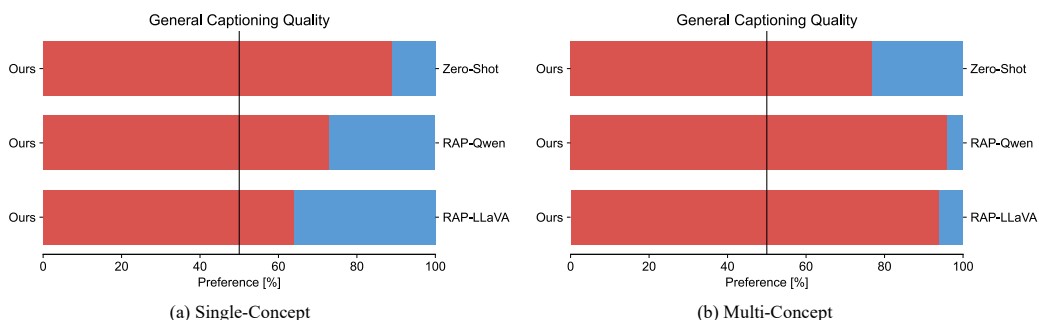

(a) Single-Concept            (b) Multi-Concept

Figure A.11: Additional preference evaluation using `Gemini-2.0-Flash` on single and multi-concept personalized image captioning tasks.

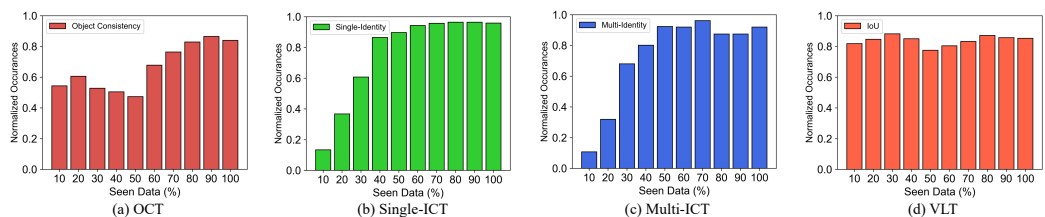

(a) OCT      (b) Single-ICT      (c) Multi-ICT      (d) VLT

Figure A.12: Distributions of mean verifiable rewards during training for each task: (a) OCT, (b–c) ICT, and (d) VLT.

## C.4 Additional Preference Evaluation Results

We acknowledge the potential bias in ChatGPT-style preference scoring, where responses that are overly polite or "sweet" may be favored. To address this issue, we enhance the evaluation in Figure 3 in our paper by incorporating results from multiple proprietary MLLMs (e.g., `Gemini 2.0 Flash`) to validate our superior performance. This multi-model evaluation strategy mitigates single-model bias and provides a more reliable assessment of overall caption quality. As shown in Figure A.11, the results are consistent with the preference rankings measured by `GPT-4o`, further reinforcing RePIC's priority in both single- and multi-concept settings.

## C.5 How Efficiently Does RL Maximize Rewards During Post-Training?

Figure A.12 illustrates how efficiently our proposed method achieves personalization of the model. To analyze this, we divide the sections with the criterion of seen data during training into bins and count the number of responses with a verifiable reward of 1 within each bin. These counts are then normalized by the total number of responses that include both rewards of 0 and 1, which we call this score as normalized occurrence. The results show a clear upward trend in performance across both OCT, single and multi-ICT, once the proportion of seen data exceeds 50%. Here, the total number of seen data is 2K. Notably, ICTs both begin with a low occurrence rate of approximately 0.2 but show a sharp emergence towards 1.0 once the seen data surpasses 50% (*i.e.*, 1K samples). These results suggest that our method effectively guides MLLM personalization in a data-efficient and effective manner, armed with our carefully designed verifiable rewards, data construction, and instruction compositions. Note, VLT shows relatively stable performance regardless of the amount of seen data.

## C.6 Can Length Regularization Reward Guides To Prolong Output Completions?

Figure A.13 presents ablation results on output completion lengths of the image captioning task across the evaluation datasets. In all cases, applying our length regularization proves as a simple yet effective strategy for increasing output lengths, consistently yielding longer completions, surpassing those generated by both zero-shot and SFT (*i.e.*, RAP-Qwen) baselines, which often generate uninformative captions such as 'This is <name>'.

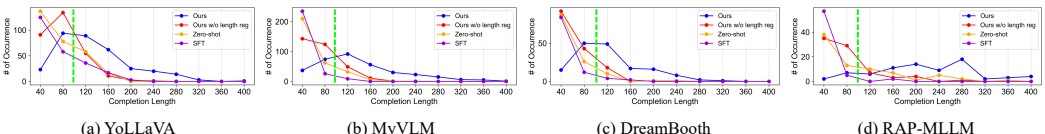

| (a) YoLLaVA | (b) MyVLM | (c) DreamBooth | (d) RAP-MLLM |
|---|---|---|---|

Figure A.13: Ablation studies on output length distributions of image captioning across single and multi-concept evaluation datasets.

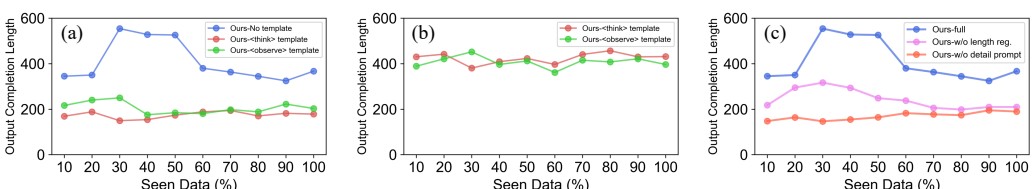

Figure A.14: Visualization of results: (a) measured output response length (*e.g.* between `<answer>` and `</answer>` tokens), (b) output length measured within the reasoning template (*e.g.* between `<think>` and `</think>` tokens), and (c) ablation studies.

## C.7 Does Reasoning Template Matter for Personalization?

We further conduct an experiment to investigate whether the reasoning templates have a meaningful impact on personalization. Specifically, we examine the effects of using special tokens and measure the output completion length when no reasoning template is used. Interestingly, as shown in Figure A.14, our ablation results reveal the following: (a) Even without a reasoning template, the model is capable of producing sufficiently long and informative answer responses. (b) In contrast, when using a reasoning template, we observe a similar tendency toward overthinking [36], which is a decline in accuracy as the average reasoning time increases, where the model focuses primarily on the reasoning process at the expense of informative answers. Thus, in the same context as our experiments in the main paper, for post-training with RL, eliminating the reasoning template contributes to enhancing the personal grounding.

## C.8 Does RePIC Enhance The General Image Captioning Ability of MLLM?

In this section, we compare the captioning performance of our proposed method with the zero-shot baseline on a general image captioning task. The evaluation does not consider both skip-retrieval and retrieval settings, as it focuses solely on captioning a single query image using general prompts without any reference images.

As a result, in Figure A.15, our method consistently generates more faithful and accurate descriptions for the image compared to the zero-shot model under general query settings. In Figure A.16, we further compare results using detailed query prompts. In this case, the results of both our method and the zero-shot model show nearly equivalent performance in caption generation. This suggests that RL-based post-training does not enhance a model's ability to perform detailed image captioning beyond what the zero-shot model can already achieve. Rather, our RL-based post-training method reinforces the frequency of more faithful and preferable captions in the output under general query prompt settings. These observations align with the results reported in concurrent studies [62, 14], and quantitative results for the preference evaluations with `GPT4o` across the MyVLM, YoLLaVa, and DreamBooth datasets are presented in Figure A.17.

Importantly, these results also demonstrate that our RePIC does not degrade the original model's general captioning capabilities after post-tuning. Unlike SFT approaches, our GRPO-based RL training maintains the model's generalization ability. This is achieved by applying KL-divergence regularization between the reference and target models during training, ensuring that the target model remains close to the reference. Thus, by maximizing a verifiable reward while preserving instruction-following ability through KL-divergence, RePIC generates the preferable personalized image captions without compromising the original model's zero-shot capabilities.

Q) Can you provide a caption of this photo?

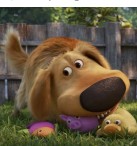

**Ours**
A cheerful animated dog with a big smile and expressive eyes lies on a grassy lawn, surrounded by colorful toys, including a purple pig and a yellow duck. The scene exudes warmth and playfulness.

**Zero-Shot**
A cheerful animated dog enjoys some playtime with colorful toys in a grassy backyard setting.

**YoLLaVA Dataset**

Q) Give a caption of the image.

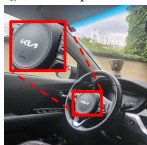

**Ours**
A playful and quirky scene inside a car, featuring a small figurine with glasses sitting on the dashboard, looking out through the windshield. The interior of the vehicle includes a modern steering wheel with a Kia logo, and the view outside shows a mix of greenery and a textured wall.

**Zero-Shot**
Steering into adventure with a quirky companion behind the wheel!

**MyVLM Dataset**

Q) Give a caption of the image.

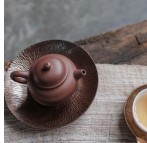

**Ours**
A serene tea setting featuring a traditional clay teapot and saucer, accompanied by a delicate cup filled with tea, all resting on a rustic wooden surface. The scene evokes a sense of calm and tradition.

**Zero-Shot**
A serene tea setting featuring a traditional clay teapot and saucer, accompanied by a delicate cup of tea on a rustic wooden surface.

**DreamBooth Dataset**

Q) Provide a caption of the given image.

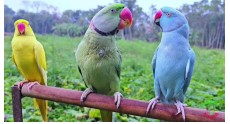

**Ours**
A vibrant scene featuring three parrots perched on a railing in a lush, green outdoor setting. The birds include a bright yellow parrot on the left, a green parrot with a pink beak in the center, and a blue parrot on the right. The background showcases a dense forest, adding depth and natural beauty to the image.

**Zero-Shot**
Three colorful parrots perched on a railing, showcasing vibrant hues against a lush green backdrop.

**RAP-MLLM Dataset**

Figure A.15: Visualization of image captioning results for general query prompts.

Q) Describe the image in detail.

**Ours**
The image features a person standing outdoors, smiling warmly at the camera. They are wearing a light blue cap with a small logo on the front and glasses. The individual is dressed in a white, short-sleeved button-up shirt with a relaxed fit. A gray jacket is casually draped over their left arm. The background showcases vibrant red flowers, likely salvia, with green foliage, suggesting a garden or park setting. The lighting appears natural, indicating daytime, and the overall atmosphere is cheerful and serene.

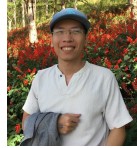

**Zero-Shot**
The image features a person standing outdoors, smiling at the camera. They are wearing a light blue cap with a small logo on the front and glasses. The individual is dressed in a white, short-sleeved, button-up shirt that appears to be made of a lightweight fabric. They have a gray jacket draped over their left arm. The background is lush with vibrant red flowers, likely salvia, and green foliage, suggesting a garden or park setting. The lighting indicates it might be daytime, possibly late afternoon given the softness of the light. The overall atmosphere is cheerful and relaxed.

**YoLLaVA Dataset**

Q) Describe the image in detail.

**Ours**
The image depicts an anime-style scene featuring two characters. On the left, there is a large, white, fluffy creature resembling a polar bear with a friendly expression, characterized by its small black eyes and a gentle smile. The creature's fur appears soft and detailed, with a mix of white and light gray tones. On the right, there is a young girl with pink hair and green eyes. She has a cheerful and content expression, with a slight blush on her cheeks. Her hair is styled in a bob cut, and she is wearing a school uniform that includes a brown jacket with gold accents, a white shirt, and a red tie. The background suggests an indoor setting, possibly a classroom or a similar environment, with wooden paneling visible behind the characters. The overall atmosphere of the image is warm and friendly.

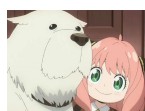

**Zero-Shot**
The image features two characters from an anime-style illustration. On the left, there is a large, fluffy white bear with a friendly expression, showing its snout and part of its body. The bear's fur appears soft and detailed, with a slightly darker shade around its mouth area. On the right, there is a young girl with vibrant pink hair and green eyes. She has a cheerful expression, with a wide smile and bright, lively eyes. Her hair is styled in a bob cut with bangs that frame her face. She wears a school uniform consisting of a brown jacket with gold trim, a white shirt, and a red ribbon or sash. The background suggests they are indoors, possibly in a hallway or room with wooden paneling. The overall tone of the image is warm and inviting, with soft lighting enhancing the characters' expressions.

**RAP-MLLM Dataset**

Figure A.16: Visualization of image captioning results for detail query prompts.

Table A.5: Visualization of Recall scores (%) for 2-concept personal grounding.

| Models | 2-Concept | | AVG |
|---|---|---|---|
| | Skip-Ret. | Ret. | |
| Ours-2K w/o multi-ICT | 43.9 | 42.7 | 43.3 |
| Ours-2K w/ multi-ICT | **98.8** | **92.7** | **95.8** |

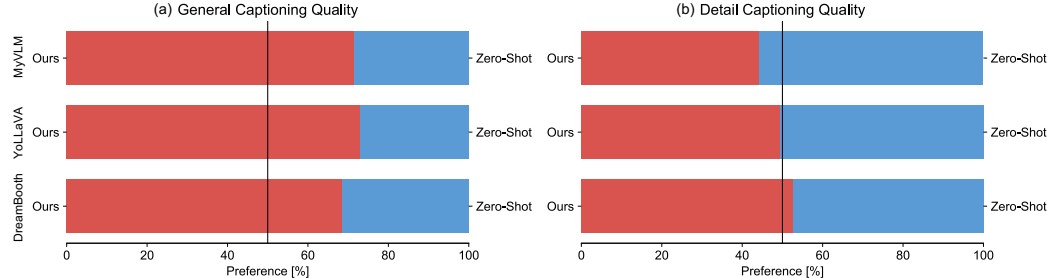

Figure A.17: Quantitative results of preference evaluations for the single-image captioning task without reference images, using (a) general query prompts and (b) detailed query prompts. Note that RePIC outperforms the zero-shot model in (a), and achieves comparable results in (b).

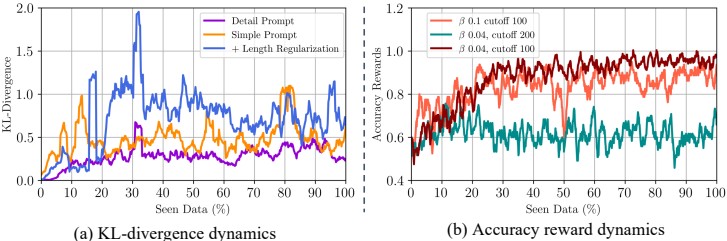

(a) KL-divergence dynamics       (b) Accuracy reward dynamics

Figure A.18: Visualization of KL-divergence and accuracy reward plots on the seen data.

## C.9 Why Multi-ICT is Necessary?

We present the recall scores in the 2-concept settings to verify the need to contain multi-ICT in the training data. As shown in Table A.5, models trained only with single-ICT fail to perform well in a multi-concept setting. This highlights the necessity of our proposed multi-ICT for improving multi-concept personal grounding performance.

## C.10 Sensitiveness of the Quality of Synthetic Data

To assess the generalizability of our method to other synthetic data variants, we incorporated the DreamBench++ dataset while keeping the data composition and hyperparameters consistent. Unlike the FLUX-based Subject200K++ used in our main experiments, DreamBench++ contains images from earlier diffusion models (e.g., SDXL [41]). As shown in Table A.6, our method achieved superior performance in both S.R and R settings. We attribute this to the higher fidelity of Subject200K++, which underwent extensive quality filtering, whereas DreamBench++ lacked sufficient refinement. We conjecture that such image fidelity highly influences our OCT reward maximization during training. Consequently, these results underscore the importance of using high-quality synthetic datasets for effective RL training.

Table A.6: Additional results of varying synthetic data in the 2-concept image captioning task.

| Models | Skip-Retrieval | | | Retrieval | | |
|---|---|---|---|---|---|---|
| | Pre. | Recall | F1 | Pre. | Recall | F1 |
| Ours w/ Subject200K++ [50] | **100** | **98.8** | **99.4** | **98.7** | **92.7** | **95.6** |
| Ours w/ DreamBench++ [39] | 100 | 89.0 | 94.2 | 97.8 | 80.5 | 88.3 |

## C.11 Analysis on Hyperparameter Sensitivity

Figure A.18 presents the results of various ablation studies. In (a), we compare three settings: using only simple prompts, incorporating detailed prompts, and further applying length regularization based

on a verifiable reward function. This reward assigns a value of 1 only when the output response length exceeds a predefined cutoff. We set the cutoff length to 100, as the average length of personal information in our database is approximately 100 tokens—roughly equivalent to at least one complete sentence. To encourage more informative image captions, we regularize the model to generate outputs of at least this length. In (b), we investigate how the expected reward changes with different values of the KL-divergence regularization weight $\beta_{\text{KL}}$. We also observe that the convergence behavior is influenced by the cutoff length used for length regularization. Our results indicate that the combination of $\beta_{\text{KL}} = 0.04$ and a cutoff length of 100 yields the best performance.

# D   Used Templates

## D.1   Evaluation Templates

In Table A.7, we present the evaluation prompts used for personalized image captioning.

Table A.7: Prompts used for evaluating the personalized image captioning experiments.

---

**Single-Concept Caption Template:**
- Give a personalized caption of this image.
- Give a caption of the image.
- Can you provide a personalized caption for this photo?
- Provide a caption of the given image.

**Multi-Concept Caption Template:**
- Give a personalized caption of this image.
- Give a personal caption of the image.
- Provide a personalized caption of the given image.

---

## D.2   Preference Evaluation Templates

The template used for our preference evaluation is shown below. Rather than favoring captions that merely duplicate retrieved content, we instructed the model to evaluate preferred captions that convey meaningful and accurate information to satisfy the following criteria:

1. **Reference Similarity:** Measures how closely the generated caption matches the retrieved reference sentence. A higher similarity indicates potential redundancy, and thus a lower preference score is assigned.

2. **Captioning Faithfulness:** Assesses how accurately the generated caption describes the visual content of the input image.

> **Preference Evaluation Template**
>
> **Retrieval-based Preference Evaluation:** You are an evaluation expert. Your task is to determine which answer best describes the given image accurately. Carefully analyze the options and select the most appropriate one as your final choice.
>
> Input: <Image>
> The name of the object in this image is: {Name}.
> The additional information for the given image is: {Info}.
> The preferable caption is one that is not merely a duplication of the given information but provides a meaningful and accurate description.
> Which one is more preferable caption to the {Name}?
> Options:
> A: {string1}
> B: {string2}
> Output the final answer by choosing one of the options with a single alphabet.
>
> Answer: A, B

## D.3 Instruction Templates

We further present the system prompts used for OCT and ICTs. In the following, in Tables A.8, A.9, and A.10, we present the full instruction templates used for OCT, ICT, and VLT, respectively. Note, we augment the instructions using GPT-4o.

> **System Prompt for OCT**
>
> As an evaluation expert, your task is to verify whether the object identified as <name> in the first image is also present in the second image. Answer with yes or no. {Question}

> **System Prompt for Single-ICT**
>
> You are a captioning expert. Your task is to generate an accurate caption for the second image while referencing the first image. Both images contain the same object. The object in the first image is named <name>. {Question}

> **System Prompt for multi-ICT**
>
> You are a captioning expert. Your task is to generate an accurate caption for the last query image while referencing the given reference images. The reference images each contain an object, named respectively as <name1>, <name2>. {Question}
>
> These are additional information about the given images except the last image: <name1>, <name2>, and <name3>. {Question}
>
> Each object in the images not including the last image has a name: <name1>, <name2>. {Question}
>
> Below is additional information about the object all images except the last one: <name1>, <name2>. {Question}

Table A.8: Instruction templates used for OCT in training data.

**Object Consistency Tuning (OCT) Template:**

- Please verify whether the objects in these pictures are the same. An object is considered the same if its consistency is maintained despite variations in lighting or pose.
- Is <name> visible in this picture?
- Is <name> in this image?
- Do you see <name> in the photo?
- Is <name> present in this photograph?
- Can you identify if <name> is captured in this picture?
- Is <name> depicted in this image?
- Does the picture feature <name>?
- Can you confirm if <name> appears in this photo?
- Is <name> included in this shot?
- Is <name> shown in this image?
- Can you tell if <name> is part of this photograph?
- Is there any sign of <name> in this picture?
- Can you detect <name> in the photo?
- Is <name> captured in this image?
- Do you recognize <name> in this picture?

Table A.9: Instruction templates used for VLT in training data.

**Visual Localization Tuning (VLT) Template:**

- Please provide the bounding box coordinate of the region this sentence describes: <name>.
- Give <name>'s bounding box in the image.
- Describe <name>'s position in the image.
- Please provide the coordinates of the bounding box for <name> in the given image.
- Specify the rectangular boundaries of <name> in the image.
- Give <name>'s position in the following image.
- Please provide <name>'s bounding coordinates in the image.
- Indicate the bounding box for <name> in the image.
- Show the bounding box for <name> in the picture.
- Specify <name>'s bounding box in the photograph.
- Mark <name>'s bounding box within the image.

Table A.10: Instruction templates used for single and multi-ICT in training data.

**Identity Consistency Tuning (ICT) Template:**

- Give a caption of the image.
- Give a personalized caption of this image.
- Provide a general caption of the image.
- Summarize the visual content of the image.
- Create a detail caption of the image.
- Offer a rich and clear interpretation of the image.
- Describe the image in detail.
- Render a summary of the photo.
- Provide a caption of the given image.
- Can you provide a personalized caption of this photo?
- Could you describe this image faithfully?
- Generate a detailed and accurate description of the image.
- Write a caption that reflects the contents and context of the image.
- Compose a meaningful caption that truly represents the image.
- Describe the image in a personalized and context-aware manner.
- Provide a natural-sounding caption that accurately conveys what is in the image.
- Craft a caption that authentically describes the scene in the image.
- Create a caption that captures the essence of the image.
- Write a caption that reflects what's visually happening in the photo.
- Generate a human-like description that accurately represents the image.
- Describe this image as if you were explaining it to a friend.
- Produce a relevant and truthful caption based on the image.
- Give a caption that matches the visual elements in the image.
- Summarize the visual content of this image in a natural way.
- Write an image-grounded caption that remains faithful to the content.
- Provide a descriptive sentence that corresponds closely to the image.

