# OpenReview forum: "RePIC: Reinforced Post-Training for Personalizing Multi-Modal Language Models"
_NeurIPS.cc/2025/Conference — NeurIPS 2025 poster_

### Official Review · Reviewer_MnDT · 2025-06-21

**Clarity:** 3
**Significance:** 2
**Originality:** 2
**Rating:** 4
**Confidence:** 2

**Summary:**

- The authors aim at identifying persons or objects by their name/identifier in images when answering a question in the sense of visual question answering. This is facilitated by presenting other images with these persons/objects as examples at inference times, either by explicit examples or by retrieving them from a source. They call this setting personalized image captioning.

- The authors present a method to retrain models (not using the term finetuning here to avoid confusion with the term supervised fine-tuning) using GRPO from one of the papers of the deepseek-AI group of researchers and compare this against supervised finetuning.
They define three types of rewards for use with GRPO supported by corresponding questions to an image.
Object consistency aims to train the model to identify an object in the target image, visual localization aims at achieving the output of bounding box coordinates with a high IoU. Identity consistency aims at the model outputting the correct identifiers for objects given by examples.

- They evaluate multiple aspects.

  - Firstly it is precision, recall and F1 for outputting the correct identifiers in single and multi-concept settings, which is a central goal of the setup of personalized image captioning. They compare here against a supervised finetuning baseline with a large dataset and a small dataset comparable to the one used for GRPO. They compare it with 1,2 and 4 identifiers present.
  - Secondly they measure chat-GPT4o preference ratings.
  - Thirdly they measure robust to providing examples of wrong objects not present in the target image.
  - Finally they perform an ablation on the three reward generating functions.

**Questions:**

- see questions in the weaknesses section:
- what is the base model used to sample outputs as reference policy ? the untuned Qwen baseline ? The model used in  training from the previous epoch ?
- it is not fully clear what the difference is between skip-retrieval and retrieval mode:
skip-retrieval presents examples of images with identifiers, query images, and example answers for the query images ? does skip-retrieval show in all of its examples only the images with the correct identifiers  (correct = present in the query image)?
[...]
- appendix:  "YOLO-World [1] is employed to detect regions of interest."
how do you achieve robustness against regions of interest which show no identifier of interest ?

addditionally:
- in the wrong GT experiments in Table 3: are all identifiers wrong or only a part of it ?

lastly:
- what is the impact or importance of or potential use cases for personalization ? To be fair, the reviewer would **not** **lower** the rating if your answer would not convince the reviewer. There is no risk of losing rating in this question. It is okay to do research on topics off the beaten path. Nevertheless the question still goes around in the mind of the reviewer.

**Ethical Concerns:**

["NO or VERY MINOR ethics concerns only"]

**Final Justification:**

The authors addressed most of my concerns. Some remain, regarding chatGPT as evaluator, the limits in novelty and the usefulness of this setup. The reviewer keeps this score, which is, nevertheless, a positive one.

**Limitations:**

- minor:
the chatgpt preference evaluation might be biases, because the provided captions might be more sweet and overly polite, and chatgpt has a tendency to prefer a conspicuously sweet style when it generates language. That should be added in the limitation section or in the appendix.

**Quality:**

3

**Strengths And Weaknesses:**

Strengths:

- Meaningful evaluation regarding precision, recall and F1 for outputting the correct identifiers.
- They show that the GRPO-based algorithm performs better in the 4-identifier setting and also when SFT does not have a much larger dataset, which brings evidence for the usefulness of the proposed approach.
- experiments with providing wrong identifiers to test robustness
- ablation study validates the usefulness of all three reward components
- evaluation templates are in the supplement

Weaknesses:

- unclear time indices t in equation 1. This is surprising because the rewards might not be properly evaluated until an answer has been fully generated. Note also that the GRPO paper does not use time indices for that reason.

- A few unclear implementation choices, namely:
- what is the base model used to sample outputs as reference policy ? the untuned Qwen baseline ? The model used in training from the previous epoch, the model lagging N=? epochs or ?
- it is not fully clear what the difference is between skip-retrieval and retrieval mode:
skip-retrieval presents examples of images with identifiers, query images, and example answers for the query images ? does skip-retrieval show in all of its examples only the images with the correct identifiers (correct = present in the query image for which the answer is evaluated)?


- appendix:  "YOLO-World [1] is employed to detect regions of interest."
how do you achieve robustness against regions of interest in the query image which show no identifier/concept of interest ?

- the chatgpt preference evaluation is considered to be of low impact, because the provided captions might be more sweet and overly polite, and chatgpt has a tendency to prefer a conspicuously sweet style when it generates language. That should be added in the limitation section.

- minor: readability: a link to details of the retrieval in the retrieval mode in appendix B.5 should be made in the main paper.
- minor: readability: a link to details of the evaluation templates in appendix D should be made in the main paper.

- minor: it would be interesting to see in 2/4 concepts how the systems behave when only a part of the identifiers is wrong.

A slightly lower score on originality because it is a straightforward application of GRPO and question templates.

Overall the reviewer is not sure to what extent personalized image captioning might have an impact, but the experiments are well-done (justifying a higher overall quality rating).
The proposed method works well with a small sample size and in settings with more identifiers.

---

> ### Author Rebuttal · Authors · 2025-07-27
>
> We sincerely thank the reviewer for the thoughtful and constructive comments on our paper. We have made every effort to respond to your feedback and faithfully reflect your suggestions. We will revise the manuscript to incorporate the following contents into the final camera-ready version.
>
> ---
>
> # Strengths
>
> We greatly appreciate that the reviewer recognized the key strengths of our work, particularly its meaningful evaluation results, the RePIC’s ability to perform well even in OOD scenarios such as the 4-identifiers (a.k.a., multi-concepts), and the validation of our three reward components through robustness testing and ablation studies.
>
> ---
> # Weaknesses
>
> **W1) Time index of GRPO in Eq (1)**
>
> In our formulation, $\pi$ denotes an autoregressive LLM that predicts the next token based on the query $q$ and all previous tokens up to timestep (a.k.a., token position) $t-1$, i.e., it can model the output $o_{t}$. While many LLM-reward frameworks assign a reward only to the final token, complicating per-token value network learning, the pioneering work of DeepSeekMath [1] (Eq. (3)) and the DAPO paper [2] (Sec. 2.2) clarify that both PPO and GRPO can define and compute token-level advantages at each timestep $t$.
>
> **W2) Definition of Base model**
>
> The model we used is Qwen2.5-VL-Instruct-7B, a vision-language model capable of processing images, text, and videos simultaneously, with enhanced instruction-following capabilities. Following the FLAN [1] paradigm, the pretrained MLLM on a large corpus was further fine-tuned using instruction datasets. We adopted this backbone because it is open-source and supports multi-image understanding, a crucial feature for MLLM personalization. We chose the 7B variant as a suitable backbone MLLM for our experiments, and the frozen copy remains fixed throughout training to perform as a reference policy while GRPO. We will add this explanation to the Appendix.
>
> **W3) Differences of skip-retrieval and retrieval settings**
>
> We apologize to the reviewer for the insufficient explanation of the skip-retrieval and retrieval settings. Although Figure 2 and lines 184–188 in the main paper, and Appendix B.5 of our paper briefly describe these settings, we clarify the differences more explicitly below:
>
> First, our database is composed of key-value pairs for all concepts, where each concept is associated with an image and corresponding textual information. Then:
>
> **1. Skip-retrieval**: In this setting, the reference samples are directly retrieved from the database by performing key-value matching based on concept names in the evaluation dataset.
>
> **2. Retrieval**: Similar to skip-retrieval, the reference image is retrieved from the database. However, as detailed in Appendix B.5, a detector first generates region proposals from the query image, and then reference images are retrieved with their texts by selecting the top-K samples from the database with high CLIP image cosine similarities.
>
> Once the reference images and texts are retrieved, they are prepended to the query prompt in an in-context learning (ICL) fashion. Importantly, no example answers for the query image are provided; MLLMs generate responses solely based on the retrieved references (images and texts) and the query image in a personalized manner. We encourage the reviewer to see Figure 2(b) in our paper for better understanding. We will incorporate this clarification into the camera-ready version of the paper.
>
> **W4) Robustness of selecting regions of interest**
>
> We sincerely appreciate the reviewer’s insightful question. As the reviewer pointed out, when performing region proposals using the YOLO-World detector, ambiguous ROIs that do not contain any identifiable concept or entity result in what we refer to as retrieval noise. Such cases impose an upper bound on the performance, and MLLM becomes inherently fragile to generate personalized captioning when the retrieved demonstrations contain content irrelevant to the query.
>
> We identify two primary causes for this issue: (1) the limited visual perception capabilities of the open-vocabulary object detection model, and (2) the absence of any relevant concept in the query image itself. We interpret the issue raised by the reviewer as corresponding to case (2). Although both scenarios present more complex challenges, our evaluation datasets do not account for the second case, as they consist primarily of object-centric images in which the target concept is clearly visible. We believe that robustness could be further improved by designing a verifiable reward that penalizes the model for generating responses based on incorrect or irrelevant information. We will incorporate this clarification into the revised manuscript.
>
> **W5) Inclusion of ChatGPT's limitations**
>
> We strongly agree with the review's opinion that ChatGPT often favors responses that are overly polite or “sweet.” To mitigate such concern, in our evaluation, we employed GPT-4o, the most capable model available, and included instructions in the system prompt to prioritize accurate and meaningful captions. However, we acknowledge that such GPT-based evaluation inherently carries intrinsic biases, including a tendency to reward politeness in output responses. We will add this limitation to our paper to clarify the potential influence of GPT-based preference evaluations. Additionally, in response to Reviewer 9oCi’s [W2], we evaluated the quality of the generated captions not only using the ChatGPT preference score but also through both reference-based and reference-free metrics. We kindly refer the reviewer to those results.
>
> **W6) Enhancing the readability of the paper**
>
> We sincerely thank the reviewer for their careful reading of our paper. As you pointed out, this is an important aspect we had not fully considered. To enhance the reader’s understanding, we will provide additional links in our paper referencing Appendix B.5 and D in the camera-ready version, possibly in the Experiments section, would be proper.
>
> **W7) More interesting results of 2/4 concept settings**
>
> We agree with the reviewer’s observation. In the revised manuscript, we will include visualizations of such cases, including instances of retrieval noise where some given identifiers are incorrect. We attribute these cases to the limited visual understanding of CLIP. To better illustrate these realistic scenarios, we will provide additional qualitative examples in the paper.
>
> **W8) Straightforward application of GRPO and question templates**
>
> While multiple factors contribute to stabilizing RL, careful reward design and data quality are among the most critical components [2]. As noted in our response to Reviewer 3m5o [L2], designing task-specific verifiable rewards is challenging, especially for downstream applications such as math. In our work, our key contributions include proposing novel verifiable rewards to guide MLLM personalization effectively. Additionally, we address the difficulty of delicately constructing datasets for MLLM personalization and identifying a “sweet spot” for stable RL training. Our results demonstrate that these efforts lead to robust performance in both single and multi-concept captioning. In future work, we aim to explore algorithmic improvements for more challenging tasks. For better understanding, please refer to the authors' responses to the reviewer v1W9 [W3], reviewer 3m5o [W1].
>
> ---
> # Questions
>
> We address the following two major questions, as most questions have been covered in our responses to the weaknesses section.
>
> **Q1) Experimental details of Table 3**
>
> In Table 3, all of the reference images contain incorrect visual concepts, while the corresponding textual information is accurate. Specifically, in this visual attack setting, we intentionally feed reference images that depict concepts different from those in the query image. This experiment is designed to evaluate whether the MLLM relies solely on textual cues and disregards the visual understanding when generating responses.
>
> **Q2) Impact or importance or potential use cases of personalization**
>
> We sincerely appreciate the reviewer’s honest and insightful question. To the best of our knowledge, MLLM personalization is still an emerging research area, and various concurrent studies are beginning to demonstrate its practical potential for the tasks of personalized image captioning, VQA, and personalized conversations, and so on. In particular, the ability to handle multi-concept inputs is becoming more essential in MLLM personalization. For example, consider a photo featuring you and your friends. In such cases, a personal multimodal agent should be able to accurately recognize and distinguish you from others to generate an appropriate personalized caption for the image.
>
> In our paper, we aimed to demonstrate the advantage of RePIC in more realistic scenarios by showing its generalizable performance even in complex group photos involving 2, 4, or even 7 individuals (see Figure A.1), including both real and AI-generated images. Furthermore, as noted in the conclusion, extending this research to other modalities such as video or audio could further enhance its real-world applicability. Lastly, exploring methods for extracting information from historical data and updating the database, as well as designing both long-term and short-term memory modules for MLLM personalization, would be valuable directions for future research. We hope this additional context helps clarify our perspective and supports the reviewer’s understanding.
>
> ---
>
> # References
>
> [1] Wei, Jason, et al. "Finetuned language models are zero-shot learners.", International Conference on Learning Representations, 2022.
>
> [2] Shao, Zhihong, et al. "Deepseekmath: Pushing the limits of mathematical reasoning in open language models." arXiv preprint
> arXiv:2402.03300 (2024).
>
> [3] Yu, Qiying, et al. "Dapo: An open-source llm reinforcement learning system at scale." arXiv preprint arXiv:2503.14476 (2025).

---

> > ### Comment · Reviewer_MnDT · 2025-08-04
> > **questions**
> >
> > regarding W1: How did you compute intermediate rewards at timesteps t different from the last token ?
> >
> > > 2. Retrieval: Similar to skip-retrieval, the reference image is retrieved from the database. However, as detailed in Appendix B.5, a detector first generates region proposals from the query image, and then reference images are retrieved with their texts by selecting the top-K samples from the database with high CLIP image cosine similarities.
> >
> > Given a region proposal, the CLIP similarities are applied between the ROI and Images ? or between the ROI and texts ? or how ?
> >
> > - The concerns on W4 (yes the reviewer has case 2 in its mind) and W5 remain.

---

> ### Author Response · Authors · 2025-08-05
>
> The authors sincerely appreciate the reviewer’s thoughtful comments and insightful questions.
>
> **C1: Intermediate rewards at different timesteps**
>
> In GRPO, all intermediate timesteps before the last token receive a uniform token-level advantage equal to the normalized final reward. As described in Sec. 4.1.2 of [2], GRPO samples a group of responses for each prompt, computes a scalar final reward $r^i$ for each response, and normalizes it by subtracting the group mean and dividing by the group standard deviation as $\hat{r}^i = \frac{r^i - mean(r)}{std(r)}$, where, $r=\{ r^1, r^2, \cdots, r^G \}$. This normalized reward $\hat{r}^i$, which serves as the token-level advantage, is uniformly assigned to every token in the corresponding response $o_i$, i.e.,
> $$ \hat{A}_t^i = \hat{r}^i \quad \forall t = 1, \ldots, |o_i|$$
> We used the same advantage calculations in our RL-based post-training, and this eliminates the need for a value network by applying the same advantage across all timesteps, enabling stable and efficient policy optimization. We will clarify this explanation in the revised manuscript.
>
> **C2: Calculating similarities in the retrieval systems**
>
> More specifically, we use an open-vocabulary detector in our retrieval step, which generates ROIs for the query image relevant to the predefined category texts specified for each concept, like "person", "dog", "toy", and so on. Then, the CLIP similarities are computed between multiple ROIs captured from the query image and the reference images in the database. Specifically, for each ROI generated by the detector, we extract its visual embedding using the CLIP image encoder and compute cosine similarity against the image embeddings of candidate reference images, also obtained using the same CLIP model. The top-K most similar reference images and their texts are then selected for MLLM demonstration. Note, we do not use reference texts in the database for the retrieval.
>
> **C3: Remaining concerns for W4 and W5**
>
> The authors appreciate the reviewer’s continued attention to the concerns regarding W4 (robustness of retrieval) and W5 (ChatGPT limitations).
>
> Regarding W4, we acknowledge the inherent limitations of the current personalized retrieval pipeline used in our work, specifically, corner cases in which the relevant concepts in the database do not appear in the query image. Although we did not consider such cases in our evaluation setting, we recognize that such scenarios can frequently occur in real-world applications. To address this, we will revise the manuscript (in the Limitations Section A.5) to include qualitative examples illustrating how our model behaves under these challenging conditions. We conjecture that such cases better highlight the limitations of current object-centric benchmarks in handling real-world complexity.
>
> For W5, we acknowledge the bias issue associated with ChatGPT-style preference scoring. To mitigate this concern raised by the reviewer, we will strengthen the evaluation results in Figure 3 by including results from multiple proprietary MLLMs (e.g., Gemini 2.5 Pro) to validate our superior performance. This multi-model evaluation strategy will help reduce single-model bias and more accurately reflect the overall quality of the generated captions.

---

> > ### Comment · Reviewer_MnDT · 2025-08-05
> > **Thank you**
> >
> > Thank you for attending to my concerns.

---

> > > ### Author Response · Authors · 2025-08-07
> > >
> > > Thank you for reviewing our rebuttal and highlighting the important point we initially missed. We’re glad to hear that our response addressed your concerns, and we will reflect your suggestions in the camera-ready version.

---

### Official Review · Reviewer_3m5o · 2025-07-02

**Clarity:** 4
**Significance:** 4
**Originality:** 4
**Rating:** 4
**Confidence:** 4

**Summary:**

This paper introduces RePIC, an innovative reinforcement learning (RL)-based post-training framework designed to enhance Multi-modal Large Language Models (MLLMs) for personalized image captioning, especially in multi-concept scenarios. The authors observe that existing supervised fine-tuning (SFT) methods struggle with faithful descriptions in complex real-world settings due to data limitations. To address this data-centric issue, RePIC leverages the Group Relative Policy Optimization (GRPO) algorithm with verifiable reward mechanisms (e.g., Object Consistency Tuning (OCT), Visual Localization Tuning (VLT), Identity Consistency Tuning (ICT)), eliminating the need for an auxiliary reward model. Experimental results clearly demonstrate RePIC's significant outperformance over existing SFT baselines in both visual recognition and personalized generation, particularly excelling in the challenging multi-concept image captioning task.

**Questions:**

1. How sensitive is the performance to the quality and diversity of the synthetic images used for OCT?

2. For Multi-ICT, how does the model handle potential ambiguities or varying relevance when associating multiple mentioned names with objects in the query image?

**Ethical Concerns:**

["NO or VERY MINOR ethics concerns only"]

**Final Justification:**

The authors have done a commendable job addressing the concerns regarding the limited technical depth of GRPO and the implementation of output length regularization. The explanations provided are both rigorous and informative:  thoughtful treatment of reward hacking in ICT through output length regularization and descriptive prompts.

These clarifications significantly improve the technical clarity of the method and demonstrate a strong grasp of RL training nuances in the context of MLLM personalization.

**Limitations:**

1. The demonstrated personalization primarily focuses on identifying and describing specific named objects, and its extendability to more nuanced personal styles or contextual understanding remains to be fully explored.

2. The direct use of GRPO without a special design for the task of this paper.

**Paper Formatting Concerns:**

No concerns

**Quality:**

4

**Strengths And Weaknesses:**

Strengths:
1）This work introduces the first RL-based post-training framework for personalized MLLM image captioning, effectively overcoming data acquisition bottlenecks inherent in traditional SFT methods.

2）Effective Solution for Multi-Concept Captioning: The paper successfully identifies and addresses the challenge of acquiring high-quality data for multi-concept image captioning, providing an RL-based solution that mitigates this limitation.

WeakNess

1) Limited Technical Depth on GRPO: The paper's explanation of GRPO's specific adaptation within the MLLM context, such as how the group of responses is generated or the reference policy established, could be more detailed.

2) The paper mentions applying "output length regularization," but more specifics on its implementation or impact on model behavior would be beneficial.

---

> ### Author Rebuttal · Authors · 2025-07-27
>
> We sincerely appreciate the reviewer’s thorough reading of our paper and the thoughtful, professional review. Reflecting on the reviewer’s comments and incorporating revisions has helped us clarify ambiguous parts of our paper and gain valuable insight into how to better highlight our contributions.
>
> # Strengths
>
> We are thrilled to introduce the first RL-based post-training framework for personalized MLLM image captioning. We are also pleased that our method effectively addresses the data acquisition bottleneck of SFT-based approaches. Furthermore, we appreciate the reviewer’s recognition that our paper successfully identifies and tackles the challenges of multi-concept captioning tasks.
>
> ---
> # Weaknesses
>
> **W1) Limited technical depth of GRPO**
>
> We appreciate the reviewer’s feedback on the limited technical depth in our current description of the GRPO. Accordingly, we provide additional explanations as follows to enhance the clarity of using GRPO.
>
> In MLLM post‑training using GRPO, the vision encoder remains frozen, and only the backbone LLM is fine‑tuned. This ensures that the visual representation (extracted by the frozen encoder) stays stable, while GRPO updates only the language model’s parameters based on RL signals.
>
> - **(1): How are group responses generated?**: For each training instance, the model generates a group of G responses $\{o_i \}^G (i=1, \cdots, G) $ by sampling from the reference policy $\pi_{\theta}$.  In our work, $\{o_i \}^G$ can be constructed with a group of personalized captions for ICT, or predicted bounding boxes for VLT, or binary yes/no responses for OCT. These responses are then converted into verifiable rewards to calculate group-relative advantages.
>
> - **(2) How are group-relative advantages normalized, and what are their merits?**: For a given set of scalar verifiable rewards, the rewards are grouped and normalized together to calculate the advantage. Given a group of rewards $r = \{r_1, r_2, \cdots, r_G\}$ obtained from each generated response, the group-relative advantage is defined as $\hat{A}_i = \frac{r_i - \mu}{\sigma}$, where $\mu$ and $\sigma$ represent the mean and standard deviation of the rewards across the group. This formulation provides a stable reward signal, even in sparse reward settings. In our work, this merit becomes prominent in single and multi-ICT rewards during post-training. In the early stages, the initial MLLM lacks personal grounding capability, resulting in sparse rewards. However, as training progresses, the rewards become richer, and we observe the emergence of personal grounding ability (see Figure A.11). We pose that this emergence is encouraged by the GRPO on learning from relatively better responses rather than relying on absolute rewards, thereby improving sample efficiency and training stability.
>
> - **(3) How is reference policy established?**: In Eq (1) of GRPO, the reference policy $\pi_\theta$ is typically established as a frozen copy of the initial pretrained MLLM, denoted as $\pi_{ref}$, and remains fixed throughout training. This frozen policy serves as a stable anchor point for regularizing the current policy $\pi_\theta$ via an explicit KL divergence penalty, which helps prevent policy drift.
>
> We will clarify these technical aspects and incorporate the additional explanations of GRPO in the camera-ready version.
>
> **W2) Details on output length regularization**
>
> We acknowledge that our previous explanation lacked clarity, and we provide a more detailed description. The output length regularization was applied exclusively to the ICT reward, not to OCT or VLT. Further, the ICT reward is granted only when the output exceeds the cutoff length. As discussed in Appendix C.2, we observed that naively maximizing the ICT reward could lead the model to exploit trivial shortcuts, such as repeatedly generating phrases like “This is \<name\>.” Since such outputs still receive a full reward of 1, this results in a reward hacking issue. To address this, we introduced a regularization strategy: if the generated output is shorter than a predefined cutoff length, the reward is set to 0, even if the name is correctly included. We empirically set this minimum length to 100, which corresponds to the lower bound of text lengths in our database. Figure A.17(b) presents a qualitative ablation across different thresholds, and we found that a value of 100 offers stable and meaningful reward signals during training. Additionally, to better encourage the model to produce more descriptive captions, we incorporated detailed prompts into the training data (e.g., “Describe the image in detail”), which promotes longer and more informative outputs. Importantly, the model’s captioning and instruction-following abilities are preserved through the KL divergence term in the GRPO loss. Consequently, as illustrated in Figure 3(b), the combination of detailed prompting and output length regularization synergistically enables the MLLM to generate the most desirable personalized captions across a variety of scenarios.
>
> ---
> # Questions
>
> **Q1) Sensitiveness of the quality of synthetic data**
>
> **Table R.1**: Additional results of varying synthetic data in 2-concept image captioning task.
>
> | **Models**         | **S.R** | Pre. | Recall | F1   | **R** | Pre. | Recall | F1   |
> |--------------------|---------|----------------|--------|------|-------|--------|--------|------|
> | Ours w/ Subject200K++ [1]    | | **100**                     | **98.8**   | **99.4** | | **98.7**                | **92.7**   | **95.6** |
> | Ours w/ DreamBench++ [2]     | | 100                     | 89.0   | 94.2 | | 97.8                | 80.5  | 88.3 |
> (S.R : Skip-retrieval, R : Retrieval)
>
> To assess the generalizability of our method to other synthetic data variants, we incorporated the DreamBench++ dataset while keeping the data composition and hyperparameters consistent. Unlike the FLUX-based Subject200K++ used in our main experiments, DreamBench++ contains images from earlier diffusion models (e.g., SDXL). As shown in Table R.1, our method achieved superior performance in both S.R and R settings. We attribute this to the higher fidelity of Subject200K++, which underwent extensive quality filtering, whereas DreamBench++ lacked sufficient refinement. We conjecture that such image fidelity highly influences our OCT reward maximization during training. Consequently, these results underscore the importance of using high-quality synthetic datasets for effective RL training.
>
> **Q2) How model handle the potential ambiguities or varying relevances?**
>
> We sincerely appreciate the insightful question. During training, Multi-ICT assigns different GT identities to manually selected, non-overlapping crops within a real image (i.e., COCO), and trains the model by rewarding the frequency of those identities appearing in the generated caption (please see Eq.(5) and Figure A.9); thus, we do not consider any potential ambiguity. On the other hand, at inference time, ambiguities may arise when the assigned reference name or textual information is incorrect or when the query image differs significantly from the reference. To test the robustness of our model, we design a visual attack scenario as described in Table 3, demonstrating that RePIC reveals robust performance against such attacks. Additionally, Figure A.4 in the Appendix shows that even when the same object (i.e., elephant) appears twice, the model can distinguish between the instances using position or contextual clues.
>
> ---
> # Limitations
>
> **L1) Extendability of RePIC on style and contextual understanding**
>
> We fully agree with the reviewer’s comment. In our paper, we only consider scenarios where visually explicit and object-centric images are provided for the personalization task. Although we briefly present a qualitative example of style personalization in Figure A.2, to the best of our knowledge, research on this domain remains very limited. We consider such extendable cases to fall under implicit personalization, where factors such as visual style, surrounding environment, background, and user intent must be interpreted contextually. We conjecture that if the visual encoder is equipped with the ability to perform such contextual understanding, it could enable more realistic and diverse personalization scenarios. This also opens up opportunities to explore new post-training methods tailored for these tasks. We will include this point in the future work section.
>
> **L2) Direct use of GRPO**
>
> We acknowledge that the novelty of RePIC may be somewhat limited due to its direct use of GRPO for post-training. However, following concurrent work, Visual-RFT [3] that uses GRPO for MLLM post-training emphasizes that careful and verifiable reward design is both important and challenging. Reviewer v1W9 also acknowledged that our work presents an innovative RL framework and leverages well-defined, verifiable reward signals to reduce data dependency. Furthermore, while current RL-based approaches have shown some effectiveness in specific tasks like math or visual recognition, our work is the first to explore RL in the context of MLLM personalization, demonstrating that it can successfully achieve generalizable captioning performance even in OOD scenarios not seen during training. For future work, we hope to explore more specialized algorithmic designs of GRPO tailored for personalization tasks targeting more complex scenarios, such as those involving 4 or more concepts.
>
> ---
> # References
>
> [1] Tan, Zhenxiong, et al. "Ominicontrol: Minimal and universal control for diffusion transformer.", The IEEE/CVF Conference on Computer Vision and Pattern Recognition, 2025.
>
> [2] Peng, Yuang, et al. "Dreambench++: A human-aligned benchmark for personalized image generation.", International Conference on Learning Representations, 2025.
>
> [3] Liu, Ziyu, et al. "Visual-rft: Visual reinforcement fine-tuning.", International Conference on Computer Vision, 2025.

---

> > ### Author Response · Authors · 2025-08-06
> >
> > We sincerely appreciate the time and effort you’ve dedicated to reviewing our submission. We would be grateful if you could kindly take a moment to review our rebuttal and share any additional feedback. Your insights are extremely valuable to us.

---

> > > ### Comment · Reviewer_3m5o · 2025-08-06
> > >
> > > Thank you for the detailed and well-structured rebuttal. The authors have done a commendable job addressing the concerns regarding the limited technical depth of GRPO and the implementation of output length regularization. I will keep my score.

---

### Official Review · Reviewer_9oCi · 2025-07-02

**Clarity:** 3
**Significance:** 2
**Originality:** 2
**Rating:** 4
**Confidence:** 4

**Summary:**

This paper introduces RePIC, an RL-based post-training framework designed to enhance the personalized image captioning abilities of MLLMs. It highlights the limitations of existing SFT-based methods in handling complex scenarios like multi-concept image captioning. RePIC incorporates three key verifiable reward components: Object Consistency, Visual Localization, and Identity Consistency, to boost the model's visual recognition and personalized generation capabilities. Experimental results demonstrate that RePIC achieves comparable performances in single-concept personalized image captioning tasks, and surpasses existing approaches in multi-concept personalized image captioning tasks, even with limited training data.

**Questions:**

Please refer to the weaknesses.

**Ethical Concerns:**

["NO or VERY MINOR ethics concerns only"]

**Final Justification:**

I appreciate the authors’ considerable effort in addressing my concerns. The rebuttal has fully resolved my doubts and, crucially, demonstrated that the paper does not simply apply GRPO to personalized captioning as a trend-chasing exercise, but rather provides solid experimental evidence of GRPO’s effectiveness for this task. Consequently, I am raising my score to 4. I encourage the authors to incorporate the additional experiments and analyses from the rebuttal into the revised version.

**Limitations:**

yes

**Paper Formatting Concerns:**

[Null]

**Quality:**

2

**Strengths And Weaknesses:**

Strengths：
1. The paper is well-written and easy to follow.
2. The framework is illustrated with clear diagrams that effectively aid comprehension.

Weaknesses：
1. Lack of Copy-Paste Analysis：The baseline achieves high performance in single-concept evaluation. This might result from the copy-paste phenomenon in MLLMs, where the model merely duplicates concept from reference concept without truly understanding the image. The paper doesn’t seem to analyze this aspect.
2. Insufficient Evaluation Metrics : The evaluation metrics only focus on whether the reference concept is correctly mentioned, which simplifies the personalized image captioning task into a retrieval problem. They fail to reflect the overall quality of the generated captions. Though Fig 4 presents two cases with satisfactory results, there’s a lack of objective quantitative metrics like BLEU, CIDEr or any other reference-free caption evaluation metrics to assess caption quality after post-training.
3. Incomplete Ablation Study: The ablation study in Table 4 doesn’t include results from the base model (without any post-training) and models using only one type of reward respectively.

---

> ### Author Rebuttal · Authors · 2025-07-27
>
> We sincerely thank the reviewer for carefully reading our paper and providing valuable suggestions for improvement. The authors are very pleased that your insightful and professional comments have significantly enhanced the quality of the current manuscript. We have carefully considered your feedback and made every effort to reflect it faithfully. We will incorporate the following changes into the final camera-ready version.
>
> ---
> # Weaknesses
>
> We have conducted extensive and quantitative evaluations to address the various weaknesses the reviewer pointed out. Please note that, due to space limitations, detailed descriptions of the evaluation metrics used will be provided in the Appendix. For single-concept evaluation, we use YoLLaVA dataset, and the reference captions are obtained via skip-retrieval from the database.
>
> **W1) Copy-and-Paste Analysis**
>
> We strongly agree with the reviewer’s concern that current MLLMs often overlook visual information and rely solely on textual cues without adequate understanding. To address this issue, we conducted the following experiments.
>
> **Table R.1**: Results of reference-based copy-and-paste analysis via measuring metric scores.
>
> | **Models**        | **BLEU (↓)** | **ROUGE-L (↓)** | **METEOR (↓)** | **SPICE (↓)** | **BERTScore (↓)** |
> |-------------------|--------------|------------------|----------------|----------------|-------------------|
> | RAP-LLaVA-210K    | 0.415        | 0.897            | 0.544          | 0.510          | 0.713             |
> | RAP-Qwen-210K     | 0.190        | 0.834            | 0.361          | 0.375          | 0.447             |
> | Qwen-2.5 VL       | 0.129        | 0.677            | 0.276          | 0.261          | 0.427             |
> | **Ours-Full-2K**  | **0.079**    | **0.578**        | **0.213**      | **0.140**      | **0.340**         |
>
> In Table R.1, we investigate the extent of copy-and-paste behavior of MLLMs. The metrics employed are commonly used in image captioning evaluation and measure the similarity between the generated output and the reference text, skip-retrieved from the database. In this context, a higher score indicates greater similarity between the two sentences, which implies a higher degree of copy-and-paste behavior. As a notable result, despite being fine-tuned on large-scale datasets, SFT-based methods are more vulnerable to the copy-and-paste problem than the zero-shot baseline. In contrast, our RL-based method consistently achieves significantly better performance across all metrics. This suggests that, unlike other SFT-based approaches that often produce templated responses without thoroughly recognizing the query image, our proposed RePIC method enables the post-tuned MLLM to generate personalized captions grounded in proper visual understanding of the query image.
>
> **Table R.2**: Results of reference-free copy-and-paste analysis via measuring text diversities.
>
> | **Prompt**           | **Models**        | **S-B (↓)** | **D-1 (↑)** | **D-2 (↑)** |
> |----------------------|-------------------|----------------|-------------|-------------|
> | **w/o detail prompt**     | RAP-LLaVA-210K    | 0.444          | 0.486       | 0.677       |
> |                     | RAP-Qwen-210K     | 0.521          | 0.383       | 0.588       |
> |                      | Qwen-2.5 VL       | 0.409          | **0.613**       | 0.542       |
> |                      | **Ours-Full-2K**  | **0.271**      | 0.539   | **0.803**   |
> | **w/ detail prompt**      | RAP-LLaVA-210K    | 0.324          | 0.483       | 0.730       |
> |                      | RAP-Qwen-210K     | 0.137          | 0.523       | 0.815       |
> |                      | Qwen-2.5 VL       | 0.359          | 0.542       | 0.782       |
> |                      | **Ours-Full-2K**  | **0.128**      | **0.624**   | **0.850**   |
> (S-B : Self-BLEU, D : Distinct-n)
>
> To further examine copy-and-paste behavior, we measured the textual diversity between output responses generated for each concept. Here, Self-BLEU (S-B) [1] treats each caption as a hypothesis and the others as references, computing BLEU scores accordingly. Distinct-n (D) [2] calculates the ratio of unique n-grams to the total number of generated tokens, offering an intuitive measure of lexical diversity. In this context, low diversity, indicated by high S-B and low D scores, suggests that the generated captions for a given concept are similar to each other, implying copy-and-paste behavior. Conversely, lower S-B and higher D scores indicate greater diversity in generated captions for each concept. Note that the “detailed prompt” refers to appending the phrase “output the response without duplicating the given information” directly to the system prompt. As shown in Table R.2, our proposed method consistently produces more diverse personalized captions compared to other methods in all settings.
>
> In summary, these results further reinforce our main claims (lines 255–256) that RePIC demonstrates strong robustness in distinguishing between objects and effectively avoids simply replicating the provided information when generating personalized captions.
>
> **W2) Image Caption Quality Measure**
>
> We conduct both reference-based and reference-free evaluations of image captioning quality on the YoLLaVA dataset. It is important to note that the employed metrics do not assess the degree of personalization achieved by the MLLM; rather, they measure only the overall quality of the generated captions for the given query images.
>
> **Table R.3**: Image caption quality evaluation results with reference captions.
>
> | **Types**        | **Metrics**       | **RAP-LLaVA** | **RAP-Qwen** | **Zero-Shot** | **Ours**     |
> |------------------|-------------------|---------------|--------------|---------------|--------------|
> | Reference-based  | BLEU (×10⁻²)      | 0.260         | 0.170        | 0.210         | **0.290**    |
> |                  | CIDEr             | 0.193         | 0.185        | **0.208**     | 0.194        |
> |                  | METEOR            | 0.242         | 0.267        | 0.271         | **0.321**    |
> |                  | SPICE             | **0.104**     | 0.084        | 0.083         | 0.086        |
> |                  | BERTScore         | **0.683**     | 0.567        | 0.523         | 0.668        |
>
> Table R.3 presents a metric-based comparison between the generated captions and GT captions (not reference text in the database) for the query image. Since no GT captions are available for the used evaluation dataset, we generated them using GPT-4o by varying the seed three times. As a result, ours show relatively similar results trends, with minor differences in the top 1–2 rankings. In detail, our proposed method achieves the best performance in BLEU and METEOR, second-best in CIDEr, SPICE, and BERTScore.
>
> **Table R.4**: Image caption quality evaluation results without reference captions.
>
> | **Types**        | **Metrics**         | **RAP-LLaVA** | **RAP-Qwen** | **Zero-Shot** | **Ours**     |
> |------------------|---------------------|---------------|--------------|---------------|--------------|
> | Reference-free   | CLIPScore (↑)       | 0.332         | 0.316        | 0.323         | **0.339**    |
> |                  | ImageReward (↑)     | -0.094        | 0.087        | **0.287**     | 0.130        |
>
> Table R.4 presents the results of image-text alignment evaluation without reference captions. CLIPScore measures the cosine similarity between image and text embeddings, while ImageReward [3] is a general-purpose reward model aligned with human preferences that evaluates image-text alignment quality. On the YoLLaVA dataset, our proposed method achieved the highest CLIPScore and the second-best ImageReward score.
>
> To sum up, these metric-based single-concept personalized captioning quality evaluation results indicate that our RePIC performs comparably to and is occasionally even preferred over zero-shot or SFT-based methods, even without considering personal grounding ability.
>
> **W3) Additional Ablation Studies**
>
> **Table R.5**: Additional ablation results of 2-concept image captioning.
>
> | **Models**         | **S.R** | Pre. | Recall | F1   | **R** | Pre. | Recall | F1   |
> |--------------------|---------|----------------|--------|------|-------|--------|--------|------|
> | Zero-Shot          | | 100                     | 75.0   | 85.7 | | 98.1                | 64.0   | 77.5 |
> | Ours only ICT      | | 100                     | 29.9   | 46.0 | | 98.0                | 25.0   | 39.5 |
> | Ours only OCT      | | 100                     | 12.8   | 22.7 | | 97.4                | 16.5   | 28.3 |
> | Ours only VLT      | | 100                     | 17.7   | 30.1 | | **98.9**                | 18.3   | 29.9 |
> | Ours 		         | | **100**                     | **98.8**   | **99.4** | | 98.7                | **92.7**   | **95.6** |
> (S.R : Skip-retrieval, R : Retrieval)
>
> We include the results of (1) zero-shot and (2) a post-tuned MLLM trained using only a single type of reward in Table R.5. All cases trained with only one type of reward show consistently low performance in personal grounding. Among them, maximizing ICT rewards yields relatively better results. Nonetheless, the results highlight that combining all proposed rewards, designed to reinforce both visual recognition and personal grounding, is essential for the success of our RePIC.
>
> ---
> # References
>
> [1] Zhu, Yaoming, et al. "Texygen: A benchmarking platform for text generation models.", The 41st international ACM SIGIR conference on research & development in information retrieval, 2018.
>
> [2] Li, Jiwei, et al. "A diversity-promoting objective function for neural conversation models." Annual Conference of the Nations of the Americas Chapter of the Association for Computational Linguistics, 2016.
>
> [3] Xu, Jiazheng, et al. "Imagereward: Learning and evaluating human preferences for text-to-image generation." Advances in Neural Information Processing Systems, 2023.

---

> > ### Author Response · Authors · 2025-08-06
> >
> > We appreciate your time and effort in reviewing our submission. We kindly request if you could please take a moment to review our rebuttal and provide any further feedback. Your insights are invaluable to us.
> >
> > Thank you

---

> > ### Comment · Reviewer_9oCi · 2025-08-06
> > **Official Comment by Reviewer 9oCi**
> >
> > Thank you for the detailed response.
> > - W1: The experiments in Tables R.1–R.2 compare the generated caption with retrieved captions from a database, but they do not test the specific copy-paste scenario I raised: when the single reference concept itself is misleading, does the model still copy it verbatim instead of grounding the description in the actual image?
> > For example, if the reference concept says “a 60-year-old man” while the image shows a young woman (as in Figure 1), does RePIC simply reproduce “60-year-old man” or does it rely on the visual content?
> > Please report the exact overlap between the model’s output and the reference concepts provided during single-concept and multi-concept evaluation, so we can verify that high scores are not driven by blind duplication of the reference text.
> >
> > - W2 & W3: The additional caption-quality and ablation results fully address my concerns; no further action is needed.

---

> ### Author Response · Authors · 2025-08-06
>
> We sincerely thank the reviewer for taking the time to carefully read and review our rebuttal. In particular, we appreciate the use of concrete examples, which helped us clearly understand the reviewer’s gist. Based on this, we would like to provide an additional response regarding the specific copy-paste scenario raised by the reviewer.
>
> ---
> **Clarifying the Understanding of the Results**
>
> Before addressing the reviewer’s concern, we would first like to clarify a potential misunderstanding. Table R.1 in our attachment presents results based on skip-retrieval of reference texts, which corresponds to the setting where ***wrong textual demonstrations*** are intentionally provided with the reference image. We sincerely apologize for not clearly communicating this in our previous response, which may have caused confusion.
>
> To elaborate, Table R.1 already evaluates the extent to which the output of RePIC overlaps with the incorrect information given in the reference texts in a single-concept setting. In other words, it assesses whether RePIC copies the misleading text verbatim rather than grounding its description in the actual visual content when the demonstration contains information that contradicts the image.
>
> As the reviewer noted, we acknowledge that our earlier explanation of Table R.2, which does not address the textual attack scenario, was not directly aligned with the reviewer’s question, and we apologize for our misunderstanding.
>
> To avoid further confusion, we would like to revise the caption of Table R.1 as follows:
>
> > → Table R.1: Single-concept copy-and-paste analysis on wrong textual demonstrations.
> | **Models**        | **BLEU (↓)** | **ROUGE-L (↓)** | **METEOR (↓)** | **SPICE (↓)** | **BERTScore (↓)** |
> |--|--------------|---|--|---|----|
> | RAP-LLaVA-210K    | 0.415 | 0.897   | 0.544  | 0.510  | 0.713   |
> | RAP-Qwen-210K     | 0.190 | 0.834    | 0.361  | 0.375  | 0.447  |
> | Qwen-2.5 VL       | 0.129   | 0.677   | 0.276  | 0.261  | 0.427  |
> | **Ours-Full-2K**  | **0.079**    | **0.578**  | **0.213**| **0.140** | **0.340** |
>
> This correction will be clearly reflected in the camera-ready version of the manuscript.
>
> Lastly, we ask for the reviewer’s understanding that per conference policy, we are unable to include any external links (e.g., introducing additional qualitative examples). Instead, we would like to showcase with a relevant example already included in our submission: in the second row of Appendix Figure A.3, the reference text includes the "black strap", but the output of RePIC does not simply duplicate this information. Rather, RePIC correctly explains the query image with "cozy yellow blanket adorned with polar bear prints" without blind duplication of the given wrong textual demonstration. This visually showcases RePIC’s robustness to textual noise and its ability to generate visually grounded captions.
>
> ---
> **C1: Reporting exact overlap scores when wrong textual demonstrations are given**
>
> For the single-concept setting raised by the reviewer, we refer to the revised Table R.1 above. In addition, we provide results for a multi-concept setting where wrong textual information for two concepts is included in the demonstration.
>
> **Table R.6**: Multi-concept copy-and-paste analysis on wrong textual demonstrations.
>
> | **Models**        | **BLEU (↓)** | **ROUGE-L (↓)** | **METEOR (↓)** | **SPICE (↓)** | **BERTScore (↓)** |
> |--|---|--|---|----|---|
> | RAP-LLaVA   	    | 0.143        | 0.649            | 0.284   | 0.335          | 0.404 |
> | RAP-Qwen     	    | 0.012        | 0.259            | 0.110          | 0.025          | 0.195 |
> | Qwen-2.5 VL       | **0.008**   | **0.080**       | **0.028**          | **0.014**    | **0.179** |
> | **Ours**  	    | 0.015    | 0.240        | 0.093      | 0.034      | 0.188  |
>
> Table R.6 presents the matching scores that quantify the overlap between the wrong textual information in the demonstrations and the generated outputs in the 2-concept setting using the RAP-MLLM dataset. In this experiment, we feed incorrect textual demonstrations for both concepts. Our method achieves the second-best performance after the zero-shot baseline and shows consistently lower copy-and-paste behavior compared to other SFT-based models.
>
> Furthermore, it is important to note that these overlap metrics do not reflect image captioning quality. Upon analyzing the outputs, we observed that the zero-shot baseline, despite producing the lowest overlap scores, frequently fails to generate meaningful captions and instead outputs only its names or irrelevant responses in this specific scenario.
>
> To sum up, these results further confirm that the strong performance of RePIC in our experiments of the main paper is not simply due to blind duplication of the reference text, but rather reflects its more robust visual recognition abilities.

---

> ### Author Response · Authors · 2025-08-07
> **Further assessment for exact overlap**
>
> Additionally, to better reflect the reviewer’s feedback, we designed and conducted a new experiment.
>
> Specifically, in both the single-concept and multi-concept settings, we manually filtered the database to include only samples containing a person. Then, following the reviewer’s suggestion, we manually applied counterfactual modifications to the reference texts for each concept (e.g., changing “a young woman” to “a 60-year-old man” or “a young man” to “a 60-year-old woman”) to create critical mismatches between the text and the image. We then measured the frequency of exact overlaps of such counterfactual modifications (i.e., "60-year-old man") between the modified text and the generated output when both the reference image and the modified text were provided as demonstrations.
>
> The frequency was calculated as the number of occurrences of the modified counterfactual phrases in the output divided by the total number of evaluation samples (i.e., occurrences / total samples). We kindly ask for your understanding that the total number of samples is relatively small, as the subset was manually filtered to include only person-related cases.
>
> The results are summarized in the table below.
>
> **Table R.7**: Frequency of responses containing counterfactual phrases.
>
> | **Models**    | **Single-Concept** | **Frequency (↓)** | **Multi-Concept** | **Frequency (↓)** |
> |---------------|--------------------|---------------|-------------------|---------------|
> | RAP-LLAVA     |                    | 51 / 90       |                   | 4 / 28        |
> | RAP-Qwen      |                    | 48 / 90       |                   | 3 / 28        |
> | Zero-Shot     |                    | 44 / 90       |                   | **0 / 28**        |
> | **Ours**      |                    | **40 / 90**   |                   | 2 / 28    |
>
> As a result, in Table R.7, our proposed method achieved the top-1 ranking in the single-concept setting and ranked second in the multi-concept setting. Notably, in the single-concept case, it exhibited a lower duplication frequency than the zero-shot baseline, and overall, it outperformed all SFT-based methods by generating fewer duplicated references across both settings. It is worth noting that in the multi-concept setting, the zero-shot model often produced useless responses that included only the name mentioned in the prompt, which explains its low frequency score.
>
> We hope this additional context helps clarify our perspective and supports the reviewer’s understanding.

---

> > ### Comment · Reviewer_9oCi · 2025-08-08
> > **Official Comment by Reviewer 9oCi**
> >
> > I appreciate the authors’ considerable effort in addressing my concerns. The rebuttal has fully resolved my doubts and, crucially, demonstrated that the paper does not simply apply GRPO to personalized captioning as a trend-chasing exercise, but rather provides solid experimental evidence of GRPO’s effectiveness for this task. Consequently, I am raising my score to 4. I encourage the authors to incorporate the additional experiments and analyses from the rebuttal into the revised version.

---

### Official Review · Reviewer_v1W9 · 2025-07-02

**Clarity:** 3
**Significance:** 2
**Originality:** 3
**Rating:** 4
**Confidence:** 3

**Summary:**

This paper proposes RePIC, an RL-based post-training framework for personalizing multi-modal language models (MLLMs) in image captioning. By integrating object consistency, visual localization, and identity consistency rewards, the method addresses the data inefficiency of supervised fine-tuning, achieving superior performance in multi-concept scenarios with only 2K training samples.

**Questions:**

Please see the points listed under Weaknesses

**Ethical Concerns:**

["NO or VERY MINOR ethics concerns only"]

**Final Justification:**

My question has been answered, and I'll keep my score.

**Limitations:**

A further consideration is the model’s performance under fine-grained visual discrepancies. When reference and query images diverge notably, the model occasionally produces descriptions that do not accurately reflect the input, hinting at limitations in the visual encoder’s robustness to complex scene variations.

**Quality:**

2

**Strengths And Weaknesses:**

**Strengths:**
1. This paper introduces an innovative reinforcement learning framework for post-training personalized image captioning in  MLLMs, marking a step forward beyond standard SFT approaches. The method is particularly compelling in how it sidesteps data dependency by leveraging well-defined, verifiable reward signals—object consistency, visual grounding, and identity alignment.
2. The framework performs strongly, especially in multi-concept scenarios. For example, it achieves a 71.0% F1 score on 4-concept skip-retrieval tasks using only 2K samples, rivalling or surpassing SFT models trained on 210K samples.
3. The paper is well-organized and easy to follow.

**Weaknesses:**
1. That said, there are a few concerns around robustness and clarity that should be addressed. The framework shows sensitivity to prompt template choices (e.g., the <observe> tag degrades performance by ~10% F1), and high ratios of ICT instructions (>31%) lead to training collapse. This suggests that the integration of components may not yet be fully stable or resilient to variations in input structure.
2.  The lack of transparency in the dataset preparation. Key aspects such as how <name> entities are labeled (manually or automatically), how multi-concept crops are selected, or whether data augmentation is applied are not clearly specified, which limits reproducibility and hinders broader adoption.
3. While GRPO intuitively appears to be a suitable optimization strategy for handling multiple identity references, the paper lacks a thorough empirical or theoretical comparison with alternative RL methods such as PPO. As a result, the specific advantages of GRPO remain somewhat speculative rather than well-substantiated.

---

> ### Author Rebuttal · Authors · 2025-07-27
>
> We thank the reviewer for your thorough reading of our paper and for providing valuable and constructive feedback. We have addressed each concern below and will incorporate all necessary revisions in the final camera-ready version.
>
> # Strength
>
> We sincerely appreciate the reviewer’s positive feedback highlighting our innovative RL framework for personalizing MLLMs, its effectiveness in addressing limitations of prior data-centric SFT methods, and its strong performance in multi-concept personalization scenarios.
>
> ---
> # Weaknesses
>
> **W1) Prompt sensitivity and training instability Issues**
>
> We appreciate the reviewer’s insight regarding the inherent instability of RL training and its sensitivity to the quality of the training dataset. RL is widely recognized as a challenging optimization paradigm [1], requiring careful reward design and delicate data curation, especially when compared to the relative simplicity of SFT.
>
> - **(1) Prompt sensitivity mitigation**: We observed that when guiding MLLMs to include \<name\> in the response, the model tends to generate generic captions without mentioning the name in the evaluation setting, if detailed prompts are not included in the training dataset. This phenomenon is especially prominent when the generated sentence becomes longer. We attribute this behavior to insufficient exploration of the model during RL training. To address this issue, we used diverse and difficult training prompts (see Tables A.4 to A.6) to prevent shortcut learning and to ensure stable personal grounding performance during evaluation.
>
> - **(2) Training instability mitigation**: In our setup, training stability is further challenged by the need to simultaneously:
> (1) preserve instruction-following capability via a KL penalty with the frozen LLM, and
> (2) maximize all different verifiable rewards (OCT, ICT, VLT).
> Despite this, we are the first to empirically identify a stable “sweet spot” in data and instruction composition and reward design that effectively balances and stabilizes the training process, which we rigorously validate in our experiments through both in-distribution and out-of-distribution experiments. We will include this important point in the Limitation section.
>
> ---
>
> **W2) Data Preparation Detail**
>
> - **(1) How \<object\> entities are labeled**: As described in Appendix B.4, we use the faker library to generate multilingual terms (e.g., person and object names) in French, Korean, Italian, Chinese, and English in an on-the-fly manner.
>
> - **(2) How multi-concept crops are selected**: For multi-concept crop images, we first select multi-image training samples (approximately 5% of training data) containing multiple crops from RAP-MLLM [2], and then manually select only those examples with non-overlapping crops and clearly visible identities. These cropped samples are used without applying additional data augmentation, such as color jittering, flip, or rotation.
>
> - **(3) Is data augmentation applied?**: Rather than applying data augmentation, we use the Subject200K+ [3] dataset, which provides pairwise high-quality synthetic images with realistic and diverse lighting and pose variations. By incorporating such a synthetic dataset, we intended to enhance the MLLM’s visual perception capabilities for handling more complex and realistic applications, such as personalized AI-generated image captioning (e.g., Figure 1, A.1-2). We will clarify these implementation details in the main paper.
>
> ---
>
> **W3) Empirical or theoretical comparison of GRPO with PPO**
>
> We provide a theoretical comparison between PPO and GRPO to explain why GRPO is a suitable algorithm for RL training in our work. Proximal Policy Optimization (PPO) [4] is a policy-based RL algorithm that stabilizes policy updates by employing a clipping mechanism, as represented in the equation below (We use the same notations in Eq. (1) of our paper).
>
> $$
> L^{PPO}(\theta) = E_{(s, a) \sim \pi_{ref}} \left[ \min \left( r_t(\theta) \hat{A}_t, \, \text{clip}(r_t(\theta), 1 - \epsilon, 1 + \epsilon) \hat{A}_t \right) \right]
> $$
>
> where $r_t(\theta) = \frac{\pi_\theta(o_{t} \mid q, o_{<t})}{\pi_{ref}(o_{t} \mid q, o_{<t})}$ represents the probability ratio between the current and reference policies, $\hat{A}_t$ is an advantage at time step t computed using the GAE (Generalized Advantage Estimation), and $\epsilon$ is a small hyperparameter controlling the clipping range. However, PPO heavily relies on absolute reward values, making it sensitive to noise or suboptimal design, and its use of a separate value function typically as large as the policy model, for advantage estimation adds significant memory and computational overhead.
>
> In contrast, GRPO is formulated as a group-level extension of PPO, incorporating per-token policy ratios, clipped advantages, and explicit KL regularization, as described in Eq. (1) of our paper. Since rewards are applied per sample, GRPO performs better with sparse rewards and stably learns from relatively better responses rather than relying on absolute rewards.
>
> In our work, GRPO offers the following advantages:
> - **(1) Effectively handling multiple identity references**: By maximizing our proposed OCT and VLT rewards, the MLLM gains enhanced visual perception and localization capabilities. Armed with these abilities, the model learns to generate faithful captions accurately grounded in multiple identities present within the query image, even without additional visual grounding signals, while maximizing single or multi-ICT rewards.
>
> - **(2) Preserving caption quality during personalization**: Instead of relying on a clipping mechanism, GRPO explicitly applies KL divergence regularization, which helps maintain instruction-following capabilities throughout the personalization process (see lines 126–128 in the Appendix).
>
> - **(3) Robust training under sparse reward conditions**: These strengths are particularly evident in single- and multi-concept ICT tasks during post-training. In the early stages, when the MLLM lacks personal grounding ability, rewards tend to be sparse and hard to learn. However, we observed that as training progresses, the personal grounding ability begins to emerge, and the reward signal becomes richer and more informative (see Figure A.11).
>
> Therefore, from those perspectives, we adopt the GRPO algorithm as a suitable strategy for our RL-based post-training because of its stability and data efficiency, even when training with a small amount of data (2K). For a more in-depth technical explanation of GRPO, please refer to the authors’ response to Reviewer 3m5o [W1]. For a detailed theoretical comparison between GRPO and PPO, please see Section 4.1.1 of [1].
>
> ---
>
> **Limitations**
>
>
> As the reviewer rightly noted, our model’s performance degrades when the reference and query images differ significantly (see Figure A.6 and Section A.5). This limitation stems from the MLLM’s difficulty in understanding high-level visual discrepancies. We agree with the reviewer that a stronger visual encoder that enables fine-grained visual perception and understanding could help mitigate this issue. Additionally, as noted in our response to Reviewer 3m5o [L1], we pose that further enhancing the visual encoder to capture nuanced personal styles and visually contextual understanding could enable the MLLM to better handle more diverse and complex personalization scenarios.
>
> ---
> # References
>
> [1] Shao, Zhihong, et al. "Deepseekmath: Pushing the limits of mathematical reasoning in open language models." arXiv preprint arXiv:2402.03300 (2024).
>
> [2] Hao, Haoran, et al. "Remember, retrieve and generate: Understanding infinite visual concepts as your personalized assistant.", The IEEE/CVF Conference on Computer Vision and Pattern Recognition, 2025.
>
> [3] Tan, Zhenxiong, et al. "Ominicontrol: Minimal and universal control for diffusion transformer.", The IEEE/CVF Conference on Computer Vision and Pattern Recognition, 2025.
>
> [4] Schulman, John, et al. "Proximal policy optimization algorithms." arXiv preprint arXiv:1707.06347 (2017).

---

> > ### Author Response · Authors · 2025-08-06
> >
> > Thank you for your time and thoughtful review of our submission. We would greatly appreciate it if you could kindly review our rebuttal and provide any further comments. Your feedback is of great importance to us. Thank you.

---

> > ### Comment · Reviewer_v1W9 · 2025-08-06
> > **General response to the authors**
> >
> > Thank you for your detailed response. Your explanations and planned revisions have successfully addressed most of my concerns.
> >
> > Regarding W1: I accept the authors' explanation. Attributing the instability of RL training to its inherent challenges and framing the discovery of a "sweet spot" for stable training as part of the contribution is a reasonable and convincing argument.
> >
> > Regarding W2 (Data preparation detail) and W3 (Justification for GRPO): The authors have also provided very thorough supplementary explanations. These key implementation details (e.g., using the faker library, manually curating multi-concept samples, and using the Subject200K+ dataset in lieu of traditional data augmentation) and the theoretical analysis of GRPO's advantages are crucial for evaluating and reproducing this work.
> >
> > My main point is that this critical information was indeed missing from the initial submission, which was the core reason for my reservations in the original review.

---

> > > ### Author Response · Authors · 2025-08-07
> > >
> > > Thank you for taking the time to review our rebuttal. We also sincerely appreciate your pointing out the crucial main point that we had overlooked.
> > >
> > > We are very glad to hear that our rebuttal has addressed your concerns, and we will make sure to incorporate all of your suggestions into the camera-ready version. Thank you again.

---

### Author Response · Authors · 2025-08-09

We introduce RePIC, the first RL-based post-training framework for personalized multimodal image captioning, which leverages GRPO with three novel verifiable rewards (object consistency, visual localization, and identity consistency) to mitigate the data-centric limitations of previous SFT-based methods and achieve strong, generalizable performance in multi-concept personalization scenarios.

---
We are encouraged by the positive feedback from all reviewers on our paper following the author–reviewer discussion period!

**Originality**: “first RL-based post-training framework for personalized MLLM” (#3m5o), “innovative” (#v1W9)

**Significance**: “successfully identifies and addresses the challenge of acquiring high-quality data” (#3m5o), “clearly demonstrate RePIC's significant outperformance” (#3m5o), “meaningful evaluation” (#MnDT)

**Technicality**: “well-defined verifiable rewards” (#v1W9), “effective solution for multi-concept captioning” (#3m5o), “solid experimental evidence” (#9oCi), "providing test robustness" (#MnDT)

**Clarity**: “well-organized and easy to follow” (#v1W9, #9oCi), “clear diagrams aid comprehension” (#9oCi), “detailed and well-structured rebuttal” (#3m5o)

**After Discussion**: “successfully addressed most of my concerns” (#v1W9), “fully resolved my doubts” (#9oCi), “detailed and well-structured rebuttal” (#3m5o), “attended to my concerns” (#MnDT)

---
We sincerely thank all reviewers for their time and constructive feedback, and especially for their active participation in the author–reviewer discussion. We also extend our gratitude to the Area Chair for their efforts. Answers to individual reviewers are addressed below each review.

---

### Decision · Program_Chairs · 2025-09-17

**Decision:**

Accept (poster)

**Comment:**

This paper introduces RePIC, an RL-based post-training framework designed to enhance the personalized image captioning abilities of MLLMs. After rebuttal, it received scores of 4444. All the reviewers are generally positive about the paper, commenting that the proposed method is the first RL-based method for the personalized image captioning tasks, with well-defined verifiable rewards. Experimental results are solid, and the paper is well-organized and easy to follow. The authors have also done a nice job during rebuttal to address the reviewers' concerns. Therefore, the AC would like to recommend acceptance of the paper.